# Tracking human skill learning with a hierarchical Bayesian sequence model

**Noémi Éltető** [1]*, **Dezső Nemeth** [2,3,4], **Karolina Janacsek** [3,5], **Peter Dayan** [1,6]

**1** Max Planck Institute for Biological Cybernetics, Tübingen, Germany, **2** Lyon Neuroscience Research Center, Université de Lyon, Lyon, France, **3** Institute of Psychology, ELTE Eötvös Loránd University, Budapest, Hungary, **4** Institute of Cognitive Neuroscience and Psychology, Research Centre for Natural Sciences, Budapest, Hungary, **5** Centre for Thinking and Learning, Institute for Lifecourse Development, Universtiy of Greenwich, London, United Kingdom, **6** University of Tübingen, Tübingen, Germany

\* noemi.elteto@tuebingen.mpg.de

**Citation:** Éltető N, Nemeth D, Janacsek K, Dayan P (2022) Tracking human skill learning with a hierarchical Bayesian sequence model. PLoS Comput Biol 18(11): e1009866. https://doi.org/10.1371/journal.pcbi.1009866

**Data Availability Statement:** The raw data, computational modeling and analysis code is publicly available at https://github.com/noemielteto/HCRP_sequence_learning. All other

## Abstract

Humans can implicitly learn complex perceptuo-motor skills over the course of large numbers of trials. This likely depends on our becoming better able to take advantage of ever richer and temporally deeper predictive relationships in the environment. Here, we offer a novel characterization of this process, fitting a non-parametric, hierarchical Bayesian sequence model to the reaction times of human participants' responses over ten sessions, each comprising thousands of trials, in a serial reaction time task involving higher-order dependencies. The model, adapted from the domain of language, forgetfully updates trial-by-trial, and seamlessly combines predictive information from shorter and longer windows onto past events, weighing the windows proportionally to their predictive power. As the model implies a posterior over window depths, we were able to determine how, and how many, previous sequence elements influenced individual participants' internal predictions, and how this changed with practice.

Already in the first session, the model showed that participants had begun to rely on two previous elements (i.e., trigrams), thereby successfully adapting to the most prominent higher-order structure in the task. The extent to which local statistical fluctuations in trigram frequency influenced participants' responses waned over subsequent sessions, as participants forgot the trigrams less and evidenced skilled performance. By the eighth session, a subset of participants shifted their prior further to consider a context deeper than two previous elements. Finally, participants showed resistance to interference and slow forgetting of the old sequence when it was changed in the final sessions. Model parameters for individual participants covaried appropriately with independent measures of working memory and error characteristics. In sum, the model offers the first principled account of the adaptive complexity and nuanced dynamics of humans' internal sequence representations during long-term implicit skill learning.

## Author summary

A central function of the brain is to predict. One challenge of prediction is that both external events and our own actions can depend on a variably deep temporal context of

relevant data are within the manuscript and its Supporting information files.

**Funding:** This research was supported by the National Brain Research Program (project 2017-1.2.1-NKP-2017-00002, PI: D.N.) and the Hungarian Scientific Research Fund (OTKA PD 124148, PI: KJ, OTKA K 128016, PI: DN). NE and PD were funded by the Max Planck Society. PD was also funded by the Alexander von Humboldt Foundation. DN was funded by the IDEXLYON Fellowship of the University of Lyon as part of the Programme Investissements d'Avenir (ANR-16-IDEX-0005). KJ was funded by the János Bolyai Research Scholarship of the Hungarian Academy of Sciences.

**Competing interests:** The authors have declared that no competing interests exist.

previous events or actions. For instance, in a short motor routine, like opening a door, our actions only depend on a few previous ones (e.g., push the handle if the key was turned). In longer routines such as coffee making, our actions require a deeper context (e.g., place the moka pot on the hob if coffee is ground, the pot is filled and closed, and the hob is on). We adopted a model from the natural language processing literature that matches humans' ability to learn variable-length relationships in sequences. This model explained the gradual emergence of more complex sequence knowledge and individual differences in an experiment where humans practiced a perceptual-motor sequence over 10 weekly sessions.

## Introduction

*"[. . .] even intuition might be reduced to mathematics".*
Isaac Asimov

The fluency and accuracy of perception and action depend critically on our preternatural ability to predict. For instance, when learning a new action routine, the steps are generated independently of each other, yielding slow and error-prone performance. Routines that are practiced extensively, like opening the front door at home, become fast, accurate, and effortless. This is because the sequence of actions comprising the routine becomes predictable via the gradual learning of dependencies among sequence elements. This sequence learning mechanism that creates skills is ubiquitous: along with its role in the genesis of fluent motor performance [1], it operates in spatio-temporal vision, assisting scene perception [2], and in the auditory domain, underlying speech perception [3–5] and speech production [6, 7].

Skill production in the form of sequence learning has been most widely studied in serial reaction time (SRT) tasks [8] in which participants are instructed to follow a repeating pattern of key presses like $A - B - A - C$. With practice, they become faster to produce the key presses obeying this sequence than those associated with a random sequence. Notably, this increase in fluency is not always accompanied by explicit knowledge of the sequence. Some participants who become faster to respond to the pattern are not able to verbally report or themselves generate the true pattern, suggesting that they learned it implicitly [9]. As such, this paradigm can capture non-intentional sequential behavior that does not require conscious awareness.

Conventional SRT tasks pose higher-order sequence learning problems. That is, the sequence elements depend on more than one previous element. In the example sequence $A - B - A - C$, whether $B$ or $C$ follows $A$ is uncertain; but $C$ follows $BA$ with certainty. That is, the first-order dependence of $C$ on $A$ is uncertain but its second-order dependence on $BA$ is certain. Learning the second-order dependencies ensures predictability, and thus fluency, for all elements of this example sequence. However, if the order of the sequence generating process is not known or instructed *a priori*, the learner has to arrive at the second-order solution by themselves. The same is true in real-life sequence learning problems. Indeed, a central challenge of sequence prediction is to determine the exploitable predictive context, the depth of which can vary from sequence to sequence, and even from element to element. Humans spontaneously adopt the depth of context appropriate to the sequence statistics [10]. Furthermore, learners in the wild accommodate substantial noise. For instance, we might have to greet a neighbor while opening our front door. Such intervening elements should be flexibly ignored in our sequence input in order to condition only on the parts of the input that belong to the door opening routine and correctly generate the next step.

The Alternating Serial Response Time (ASRT) task [11] was developed to study higher-order sequence learning in the face of noise. The paradigm is identical to that of the SRT but the sequence of key presses is predictable only on every alternate trial. Participants gradually respond more quickly on predictable trials, presumably because they learn to exploit a sufficiently deep context—that is to say, they form larger context-action chunks. However, due to the probabilistic sequence, participants' knowledge in the ASRT is completely implicit [12], as opposed to the mixed explicit-implicit knowledge that is typically exhibited in the SRT. Therefore, one can assume that the response times in the ASRT are predominantly influenced by the probability of the upcoming elements and not by other, explicit, strategies. We used the ASRT to study how humans adapt implicitly to higher-order structure in a noisy sequence—providing unique insight into the long-term learning of a complex skill.

Since Shannon [13], so-called $n$-gram models have been a conventional approach to modeling non-deterministic higher-order sequential dependencies. An $n$-gram model learns to predict the next element given the previous $n − 1$ elements. For instance, a 3-gram (or trigram) model predicts an element given two previous elements. In essence, an $n$-gram is a *chunk* of $n$ adjacent elements, and we use the terms interchangeably. One major limitation of such models is that the number of $n$-grams grows exponentially as a function of their size $n$. Thus, acquiring or storing an $n$-gram table becomes statistically and computationally infeasible, respectively, even at moderate values of $n$. Critically, a simple $n$-gram model fails to exploit the typically hierarchical nature of chunks: i.e., that a chunk 'inherits' predictive power from its suffix. For instance, in the speech prediction example, given a context of 'in California, San', the most distant word 'in' is weakly predictive, while 'California' and 'San' are strongly predictive of 'Francisco'. The entire context 'in California, San' inherits most of its predictive power from the shallower context 'California, San'. Similarly, action chunks underlying our motor skills, like opening a door, are often embedded into, and interrupted by, previously unseen or irrelevant actions. Humans appear capable of exploiting the hierarchical statistical structure of sequences by down-weighting, or ignoring, parts of the context that have not convincingly been observed to be predictive.

Teh [14] suggested a Bayesian non-parametric extension of $n$-gram models as a principled machine learning solution to both the problem of complexity and hierarchical context weighting. This model builds structure on the fly as evidence accumulates, extending from $(n − 1)$- to $n$-gram dependencies according to the observed statistics. Thus, it flexibly reduces to a unigram model if no chunk is present, or builds the bigram, trigram, etc. levels if appropriate. For prediction, it smooths over all chunks that are consistent with the available context, proportional to their prior evidence. This model was originally suggested as a language model; here, we consider its use for a more general cognitive contextual sequence learning problem.

In our experiment, participants practiced the same visuo-motor second-order sequence in the ASRT task for 8 long sessions, each separated by a week. In two subsequent sessions, the sequence was changed in order to test participants' resistance to interference. We tracked the evolution of sequence knowledge using the Bayesian non-parametric sequence model, capturing representational dynamics and sensitivity to local statistical fluctuations by adapting it to learn online and be suitably forgetful. We fitted the sequence model to participants' response times assuming that faster responses reflect more certain expectations. We show how shifting their priors over the predictive contexts allowed participants to grow and refine their internal sequence representations week by week. Already in the first session, participants began to rely on two previous elements for prediction, thereby successfully adapting to the main task structure. However, at this early stage, trigram recency influenced their responses, as captured by the forgetting mechanism of our model. With training, trigram forgetting was reduced, giving rise to robustness against local statistical fluctuations. Thus, our model reduced to a simple,

stationary trigram model. However, by the last training session, we observed that a subset of participants shifted their prior further to consider a context even deeper than two previous elements. The fitted parameter values guiding higher-order sequence learning were correlated with independently measured working memory scores. Finally, reduced chunk forgetting predicted the resistance to interference in the last two sessions.

# 1 Methods

## 1.1 Ethics statement

All participants provided written informed consent before enrollment and received course credits for taking part in the experiment. The study was approved by the United Ethical Review Committee for Research in Psychology (EPKEB) in Hungary (Approval number: 30/2012) and by the research ethics committee of Eötvös Loránd University, Budapest, Hungary. The study was conducted in accordance with the Declaration of Helsinki.

## 1.2 Experiment

A detailed description of the task, procedure, and participants can be found in [15] where this data was first published. In brief, we tested participants' long-term sequence learning on a serial reaction time task with second-order dependence (the Alternating Serial Reaction Time task or ASRT [11]). On each trial, a cue appeared in one of four equally spaced, horizontally arranged locations. Participants had to press a corresponding key as accurately and quickly as possible. The next trial started 120ms after the response (Fig 1a). An eight-element second-order sequence dictated the cue locations, i.e. the sequence elements. In this, four deterministic

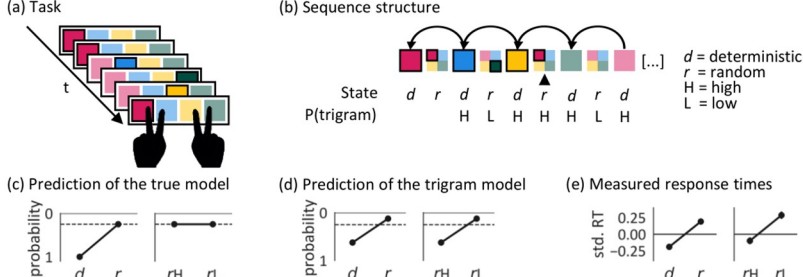

**Fig 1. The Alternating Serial Reaction Time (ASRT) task with second-order dependence structure.** (a) Participants had to press the key corresponding to the current sequence element (i.e. cue location) on the screen as accurately and quickly as possible, using the index and middle fingers of both hands. In the display, the possible locations were outlined in black and the cue always looked the same, fill color and saturation are only used here for explanatory purposes. (b) The structure of the example sequence segment in (a). Color saturation and outline indicate the element that was presented on a trial. The vertical arrow indicates the current trial. The task was generated from an eight element second-order sequence where every second element was deterministic and the elements in between were random. The deterministic components in this example are: red-blue-yellow-green. The element on any random trial (including the current one) is unpredictable. However, this current trial happens to mimic the deterministic second-order dependence where green is followed by red after a gap of one trial, making it a high probability trigram trial (H). The other random elements were associated with lower probability trigrams (L). (c) Under the true generative model, when in a random state, high-probability trigrams (rH) and low-probability trigrams (rL) are equally unexpected. (d) A learner who can pick up second-order dependencies, but who is agnostic to the higher-order alternating state structure, would expect rH more than rL. (e) In the last training session (session 8; after more than 14,000 trials), participants responded faster to deterministic than random trials, suggesting that they learned to predict the upcoming element. They also responded quickly even on random trials if those happened to complete a high probability trigram (rH). The y axis shows the standardised reaction time (RT) averaged over the different trial types on the last session of learning. The error bars indicate the 95% CI.

states, each associated with a unique element, were interleaved with four random states, which produced the four elements with equal probabilities.

This second-order rule implies that a deterministic element is predictable from the element two time steps ago. If one ignores the deterministic/random state of the alternating sequence, this also means that some trigrams (i.e., sequences of three elements) have high probabilities. Such trigrams can also arise, by chance, in the random states (Fig 1b), and so allowed [11] a test of whether participants had learned the global alternating rule in the task, in which case any element in a random state would be unexpected (by the time the state had been inferred), or if they had instead merely learned local dependencies or frequent chunks, in which cases a random state that happened to complete a (so-called random) high frequency trigram would also be expected. The excess speed of responses to the final elements of random high-frequency trigrams compared to random low-frequency trigrams shown in Fig 1c suggests chunk learning. Learning was purely implicit in this task, as none of the participants could verbalize or reproduce the sequence structure in the debriefing at the end of the experiment.

Participants completed nine sessions of 2125 trials each, and a tenth session of 1700 trials, each separated by a week. For each participant, one of the six unique permutations of the deterministic second-order sequences was selected in a pseudo-random manner and the same sequence was practiced for 8 (training) sessions. On session 9, unbeknownst to the participants, two elements in the deterministic second-order sequence were swapped and thus all but one of the second-order pairs were changed. Over four, 425 trial, epochs of session 10, old and new sequences alternated according to *old → new → old → new*. We refer to sessions 9 and 10 as interference sessions. Of the 32 participants, we analysed data from the 25 (22 females and 3 males; $M_{age}$ = 20.4 years, $SD_{age}$ = 1.0 years) who completed all ten sessions.

## 1.3 Modeling strategy

We assume that a learner predicts the probabilities of successive elements in the sequence, and that the response alacrity for pressing key $k$ on trial $t$ ($\tau_t$) scales logarithmically with the probability the model awards to that $k$. Thus high-probability responses are the fastest [16] due to being most expected (recent neural evidence in [17]) (Fig 2 upper box). Note that fitting to the participants' responses rather than the actual events allows us to make inferences about their

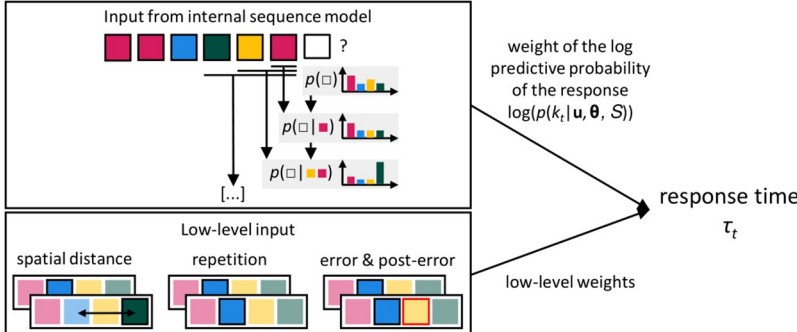

**Fig 2. Modeling strategy.** We adopted a model-based approach, fitting the hyperparameters $\theta$ of an internal sequence model (upper box), together with low level effects (the spatial distance between subsequent response locations, response repetition, error and post-error trials; lower box) to participants' response times. The contribution of the sequence model is the scaled log of the predictive probability of each key press $k$ (one of the four keys, marked as transparent square), given the context **u** (previous events, marked as a string of colored squares). The sequence model makes predictions by flexibly combining information from deepening windows onto the past, considering fewer or more previous stimuli.

internal sequence model from their errors as well as their correct responses. For this, we make the assumption that errors reflect not only less caution but also expectations (as captured by lower response threshold and biased starting point in evidence accumulation models, i.e. [18]). Indeed, individuals are prone to respond according to their internal predictions, even if these do not match the actual upcoming elements [19]. In the ASRT, where the current element is already presented at the time of the response, supposedly a *conflict* arises between the instructed response and the error response. However, the nature of the conflict is not within the scope of this study.

The learner faces the problem of finding a model that considers a wide enough context of predictive past elements, whilst not suffering from combinatorial explosion and overfitting by considering too many past elements that are redundant for prediction. The solution we consider in this paper is a Bayesian nonparametric *n*-gram model [14]. In a nutshell, the model combines the predictive information from progressively deeper windows onto the past: no previous element; one previous element; two previous elements etc., corresponding to the unigram, bigram, trigram, etc., levels. The hierarchies in the model provide a principled way of combining information: a deeper window 'inherits' evidence strength from the shallower windows that it contains.

Teh [14] employed an offline algorithm to model a given static sequence such as a text corpus. However, the model can be fitted in an online sequential fashion instead, updating the beliefs at each observation and using the updated model for predicting the next observation. This captures representational dynamics: more complex models are employed as the data to justify them accumulates. We hypothesized that humans build their internal sequence representation in a similar way, starting with learning short-range dependencies and gradually adding long-range dependencies if they are indeed significantly present in the data. Therefore, we adopted this model to serve as an *internal* sequence model.

In order to isolate the effect of the internal sequence prediction on reaction times (RTs), we controlled for low-level effects that exert significant influence in serial reaction time tasks (S2 Table). These included the spatial distance between subsequent elements, repetition of the same location, errors, and post-error trials (Fig 2 lower box). Thus, we performed a single, integrated fit of the free parameters of the sequence predictor and the low-level effects to the recorded RTs.

## 1.4 The internal sequence model

The model from [14] infers a nested predictive context of sequence elements, probabilistically choosing the level of the nesting for a particular prediction based on priors and data. Since we treat it as an *internal* and *subjective* rather than a normative and objective model (fitting parameters to participants' reaction times rather than to the actual structure of the second order sequence), we can infer how much sequence information participants used for adapting their behavior. Using online model fitting, we capture the dynamic, trial-by-trial refinement of individuals' sequence representation.

At the core of the model is the Dirichlet process (DP) mixture [20]. This places a prior on the unigram probabilities $\mathbf{G}(k)$ of key presses $k$:

$$k \sim \mathbf{G} \sim DP(\alpha, \mathbf{H}) \tag{1}$$

where $\alpha$ is called a strength parameter, and $\mathbf{H}$ is a base probability distribution. The DP is a prior over how to cluster data points together, with the strength parameter determining the propensity for co-affiliation. In our case, each cluster is labeled by $k$. Thus, the DP expresses a prior over of a future key press, given a history of key presses. The base distribution determines

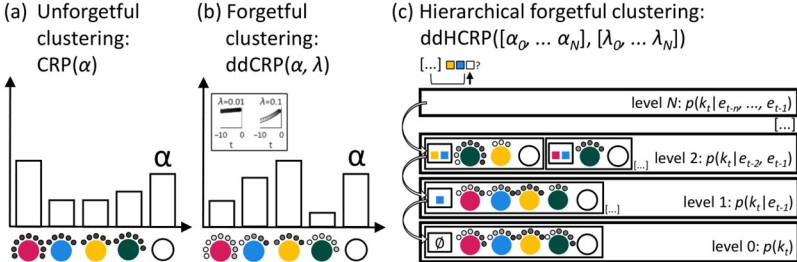

**Fig 3. Treating the sequence learning problem as an hierarchical nonparametric clustering problem.** (a) The traditional, unforgetful Chinese restaurant process (CRP) is a nonparametric Bayesian model where the probability that a new observation belongs to an existing cluster or a new one is determined by the cluster sizes and the strength parameter $\alpha$. In the metaphor, the new customer (new observation; see the terminology in Table 1; shown as black dots) sits at one of the existing tables (clusters labeled by key press identity, e.g., 'response to left side of the screen'; shown as colored circles) or opens up a new table (shown as open circle) with probabilities proportional to the number of customers sitting at the tables and $\alpha$. Here, the most likely next response would be of the type *pink*. (b) The distance-dependent or 'forgetful' Chinese restaurant process (ddCRP) is governed by a distance metric, according to the 'close together sit together' principle. In our case, the customers are subject to exponential decay with rate $\lambda$, as shown in the inset (and illustrated by the grey colours of the customers). Even though the same number of customers sit at the tables as in (a), this time the predictive probability of a *yellow* response is highest because most of the recent responses were *yellow*. (c) In the distance-dependent hierarchical Chinese restaurant process (HCRP), restaurants are labeled by the context of some number of preceding events and are organized hierarchically such that restaurants with the longest context are on top. Thus, each restaurant models the key press of the participant at time point $t$, $k_t$, given a context of $n$ events ($e_{t-n}, \ldots e_{t-1}$). A new customer arrives first to the topmost restaurant that corresponds to its context in the data (in the example, the customer is bound to visit the restaurant labeled by the context 'yellow-blue' when he arrives to level 2). If it opens up a new table, it also backs off to the restaurant corresponding to the context one element shorter (in the example, to the restaurant labeled by the context 'blue').

the prior probabilities of cluster labels $k$. In our case, **H** is uniform, expressing that we have no information about the probabilities of the key presses before observing the data.

A commonly used procedure for sampling from **G** is the Chinese restaurant process (CRP) [21]. The CRP is both a generative and recognition procedure as we explain below. In the CRP metaphor, a new customer either joins an occupied table with a probability proportional to the number of customers at that table (making the model affiliative) or opens up a new one (making the model infinite), with probability proportional to parameter $\alpha$ (Fig 3a). Each table corresponds to a cluster and, in our case, is labeled by a key press (e.g., 'key 1', marked as colors in Fig 3a). The customers correspond to observations of the certain key press (see the summary of the terminology in Table 1). In the recognition procedure, we treat the response as given and the customer is bound to sit at or open a table with the corresponding label, i.e. the probability of the given key press belonging to each cluster is computed. The fact that the same key press can be the label of different clusters reflects that the same response could arise from different latent causes, e.g., different contexts. In the generative model, the probabilities of sitting at or opening tables that share the label are summed, i.e. predicts how likely, on average, each key press is. The strength parameter $\alpha$ controls the expected number of clusters: the higher $\alpha$ is, the more prone

**Table 1. Terminology of the hierarchical Chinese restaurant process mapped onto the experimental measures of the current study.**

| ddHCRP metaphor | Experimental measures |
|---|---|
| customer | observation at $t$ |
| table | subset of observations up to $t - 1$ |
| dish | label of the observation at $t$ (key press $k_t$ or event $e_t$) |
| restaurant on level $n$ | context of $n$ previous events ($\mathbf{e}_{t-n:t-1}$) |

customers will be to open their own cluster. The resulting seating arrangement **S** is a sample of **G**. Since in the generative procedure the labels of new tables are sampled independently from **H**, a high strength value $\alpha$ will cause **G** to resemble **H** (hence, enhancing the 'strength of **H**').

The affiliative process gives rise to the 'rich gets richer' property where clusters with more data points attract other data points more strongly. This captures, for instance, the fact that the more often we gave a response in the past, the more likely we are to repeat the same response in the future [22]. However, these clusters would ultimately grow without bound. Since our participants are forgetful and might be sensitive to local statistical fluctuations, we used a variant called the distance-dependent CRP (ddCRP) [23] (Fig 3b). Here, affiliation among customers decreases as a function of sequential distances **D**. **D** is the set of distances between all customers that form sequential pairs, measured in the number of *past* trials relative to the current customer. We set the affiliation decrease to be exponential, with rate λ, crudely modeling human forgetting curves.

This yields the distance-dependent Dirichlet process (ddDP) prior:

$$k_t \sim G_t \sim ddDP(\alpha, \lambda, \mathbf{H}, \mathbf{D}) \tag{2}$$

So far, we have a suitable model of (potentially nonstationary) marginal probabilities of key presses that corresponds to a unigram model. We can use this building block to represent the key press probabilities at time point $t$ conditioned on a context of $n$ preceding key press instructions or events $\mathbf{u}_t = (e_{t-n}, \ldots e_{t-1})$ hierarchically. The Dirichlet process prior in Eq 2 can be modified to give a distance-dependent hierarchical Dirichlet process (ddHDP) prior:

$$k_t \sim G_{\mathbf{u}_t} \sim ddHDP(\alpha_{|\mathbf{u}_t|}, \lambda_{|\mathbf{u}_t|}, \mathbf{G}_{\pi(\mathbf{u}_t)}, \mathbf{D}) \tag{3}$$

where $\pi(\mathbf{u}_t)$ is the suffix of the context $\mathbf{u}_t$ containing all but the earliest event. Both the strength and decay constant are functions of the length $|\mathbf{u}_t|$ of the context. Crucially, instead of an uninformed base distribution, we have $\mathbf{G}_{\pi(\mathbf{u}_t)}$, the vector of probabilities of the current events given all but the earliest event in the context. Of course, $\mathbf{G}_{\pi(\mathbf{u}_t)}$ also has to be estimated. We do this recursively, by applying Eq 3 to the distribution given the shallower context. We continue the recursion until the context is 'emptied out' and the probability of the next response given no previous elements is a DP with an uninformed base distribution. This hierarchical procedure ensures that context information is weighted proportionally to predictive power. Consider an example where participants are always instructed to respond 'red' after having seen 'green—yellow'. Then, irrespectively of elements preceding 'green', the response should be unchanged. Consider that a novel element 'cyan' was inserted and the participant's full context contains 'cyan—green—yellow'. According to Eq 3, the probability of the next response being 'red' given the full context 'cyan—green—yellow' will depend on the probability of 'red' given the shallower context of the two previous elements 'green—yellow'—the actually predictive context. The earliest element in the context—'cyan'—is redundant to prediction and the probability distribution over the next response given the longer context will strongly resemble the probability distribution given the shorter, useful context. Note that we use the completely novel element 'cyan' to illustrate the extreme case of an unpredictive element that should be ignored. However, the principle applies to weakly predictive contexts that should be proportionally down-weighted.

We represent the HDP with the distance-dependent hierarchical Chinese restaurant process (HCRP) (Fig 3c). Here, we have Chinese restaurants on $N+1$ levels, each level modeling the probability distribution over responses given a context of $n$ previous events. At each level $n$, we have a potentially unbounded number of restaurants identified by a context **u**, meaning that a customer $k_t$ can only ever access the restaurant if $\mathbf{u}_t$ is part of their context in the data. The 'reliance' of deeper contexts on shallower ones is realised by a back-off procedure (C

Algorithm in S1 Appendix). A customer visits the topmost level first, particularly the restaurant corresponding to its context of length $N$ in the data. With probabilities proportional to the recency of customers and $\alpha_N$, the customer either joins previous customers or opens up a new table, creating a new cluster. In the latter case, the customer also 'backs off' to the restaurant below, identified with a context that is one event shallower, where the same seating evaluation takes place. This may be repeated until the customer reaches the level of empty context. This induces a process where the most predictive segment of a long context will contribute the most to the prediction of the next key press $k_t$, and superfluous context is organically overlooked. The most likely next $k$, given the context, is the $k$ that the participant recently chose in the same context *or* a shallower segment of it.

A relevant property of the HCRP is that the 'rich gets richer' effect will generalise across contexts that share the same suffix. For instance, in case of many recent observations of the response 'red' in the context 'green—yellow', the longer, previously seen context 'green—green—yellow' and even a previously unseen context 'cyan—green—yellow' will also be associated with increased likelihoods of 'red'. This is because 'red' is likely under the common suffix of both contexts. This property is desirable for the prediction of behavior, as individuals are expected to generalize their chunk inventory to previously unseen wider contexts [24, 25].

The $\boldsymbol{\alpha}$ and $\lambda$ parameters control memory over two timescales; the former controls short-term memory by tracking the use of the current context and the latter controls long-term memory by determining the longevity of previous chunks of context and response. Short-term memory acts as activated long-term memory [26]. That is, the context of a few previous events is a pointer to responses made in the same context, stored on the long-term. Since the $\alpha$ and $\lambda$ parameters are specific to the hierarchy levels and are inferred from the data, the sequence learning problem is cast as learning a prior over 'What should I remember?', as in other approaches to representation learning [27].

To a learner that knows the alternating structure and the current state of the ASRT, any context longer than two previous events is superfluous, due to the second-order dependencies (Fig 1c). However, if the learner is agnostic to the alternating structure (Fig 1d) then no context can be deemed superfluous, as longer contexts enable the implicit tracking of the sequence state. Indeed, pure sequence prediction performance increases with the number of levels in the HCRP hierarchy and with lower values of $\alpha$ (S2 Fig). Similarly, long-term memory is beneficial in the training sessions, as it allows for a better estimation of the stationary chunk distribution and provides resistance to local statistical fluctuations. However, human learners are solving the task under resource constraints, motivating them to increase the complexity of their representations only to a level that enables *good enough* performance [28]. Within our framework, a parsimonious sequence representation is 'carved out' by learning to ignore dispensable context and enhancing the memory of previous observations in the necessary context.

## 1.5 Parameter fitting

Given the sequence presented to the participants, their responses, and response times, we are interested in finding the parameter values of the low-level effects and the internal sequence model that most likely generated the behavior. We assumed that the likelihood of the log response times was a Gaussian distribution with a mean value of the log response times predicted by the full model. We performed approximate Bayesian computation (ABC) to approximate the maximum a posteriori values of the parameters of interest:

$$\underset{\boldsymbol{\theta}, \boldsymbol{\rho}}{\operatorname{argmax}} P(\boldsymbol{\theta}|\mathbf{e}, \mathbf{k}, \boldsymbol{\tau}, \sigma) \tag{4}$$

where $\boldsymbol{\theta}$ is the parameter vector of the HCRP comprising the strength parameters $\boldsymbol{\alpha}$ and

forgetting rate parameters $\boldsymbol{\lambda}$; $\boldsymbol{\rho}$ is the vector of response parameters, including the weights of the low-level effects, the weight of the HCRP prediction, and the response noise; $\mathbf{e}$ is the sequence of events (mapping onto required responses); and $\mathbf{k}$ is the sequence of key presses (actual responses); $\boldsymbol{\tau}$ is the sequence of response times; and $\sigma$ is the Gaussian noise of the response time.

As a first step of our ABC procedure, we parsed $\mathbf{e}$ and $\mathbf{k}$ chronologically, such that the probability of a key press at $t$ was influenced by observations up to $t − 1$, modeling sequential information accumulation (A Algorithm in S1 Appendix). At each time step, the HCRP operated as both the generative and recognition model, based on the same hierarchical back-off scheme. On trial $t$, we evaluated the probability of seating a new customer to a table serving dish $k_t$, according to the generative process (B Algorithm in S1 Appendix). This corresponded to computing the predictive probability $p(k_t)$ of the participant's response. Then, the seating arrangement was updated by seating a customer to a table serving the dish $e_t$, according to the recognition process (C Algorithm in S1 Appendix). This corresponded to updating the participant's internal model with the event $e_t$, that is, the required response (which, in the case of erroneous responses, was different from the actual response $k_t$). As such, we modeled the generative process of the *actual* responses $\mathbf{k}$ as a function of having learnt the *required* responses $\mathbf{e}$. Note that in the parsing procedure, the seating arrangement was only updated with the current customer—backtracking (i.e. re-seating old customers) was not possible. This models online, trial-by-trial learning. We parsed the sequence five times to generate five seating arrangements. $p(k_t)$ was averaged over the five seating arrangement samples (yielding a relatively high-precision estimate of $p(k_t)$; see A Fig in S2 Appendix).

The log predictive probability $\log(p(k_t))$ of the actual response was assumed to be linearly related to $\tau$, higher surprise causing slower responses [16]. We mapped $\log(p(k_t))$ to $\boldsymbol{\tau}_{predicted}$ using the response parameters $\boldsymbol{\rho}$. Then we computed the Gaussian densities $p(\boldsymbol{\tau}|\boldsymbol{\tau}_{predicted}, \sigma)$. The goal was to find $\boldsymbol{\theta}$ and $\boldsymbol{\rho}$ that maximize the product of these densities, that is, the likelihood of the measured response latencies to the sequence elements. In order to approximate $\boldsymbol{\theta}$ and $\boldsymbol{\rho}$ that maximize this likelihood, we performed random search in the space of $\boldsymbol{\theta}$ for 1000 iterations (for the convergence of the random search procedure, see A Text in S2 Appendix). In each iteration, we fitted $\boldsymbol{\rho}$ using OLS (thus, $\boldsymbol{\rho}$ was a deterministic function of $\boldsymbol{\theta}$). We repeated the search procedure 10 times, yielding 10 samples from the posterior distribution, and $\boldsymbol{\theta}$ associated with the highest likelihood of participants' responses was chosen as the MAP estimate of the hypermarameters.

We reran the ABC on consecutive data bins (sessions or within-session epochs when higher resolution is justified) to track potential shifts in the posteriors over practice. In the first five bins (five epochs of session 1), the prior was uninformed (S1 Table, left). In each successive bin, the prior was informed by both the fitted *hyperparameters* and *learned* parameters from the previous bins. For the hyperparameters $\boldsymbol{\theta}$, we used a Gaussian prior with a mean of the MAP values of the hyperparameters from the previous bin and a fixed variance, truncated at the boundaries of the uniform prior for session 1 (S1 Table, right). The *learned* parameter values, that is, the seating arrangements $\mathbf{S}$ accumulated across all previous bins were carried over. The 'heredity' of the seating arrangements modeled continual learning. Nevertheless, changes in $\boldsymbol{\theta}$ caused the same sequence information to be weighted differently. For instance, if $\boldsymbol{\lambda}$ decreased, old instances of chunks that were previously uninfluential, became more influential.

For later model evaluation, we held out the middle segment of reaction time data from a session (the central 255 trials) or epoch (the central 85 trials). Middle segments were chosen in order to ensure a more representative test data, as the beginning and end of a session can be affected by warm-up [29] and fatigue [30] effects, respectively. The HCRP parsed the entire $\mathbf{e}$

in order to contain sequence knowledge that could explain $\tau$ on later segments. But, importantly, the middle segment of $\tau$ was not used for computing the posterior probabilities of $\theta$ and $\rho$. Therefore, all predictions reflected all previous observations and only predictions of the training data were used to optimise the hyperparameters. Holding out the responses but not the observations, instead of completely held-out data as typical in machine learning, was essential to provide the model with the right context for predicting behavior without contaminating it with test behavior.

## 2 Results

### 2.1 Practice-related changes in the low-level response model

In Fig 4 we show fitted values of both the response parameters $\rho$ and the internal sequence model hyperparameters $\theta$. In general, responses were faster if they were repetitions of the previous responses and and if they were erroneous (Fig 4a, negative coefficients for 'repetition' and 'error'). On the other hand, slowing as a function of spatial distance from the previous response location and post-error slowing was observed (positive coefficients for 'spatial distance' and 'post-error'). Since the sequence prediction coefficient (i.e. the weight of the HCRP) expresses the effect of surprise and the sequence was not completely predictable, the coefficient was generally above zero.

Parameter dynamics were already evident in the first session. To test practice-related changes, we conducted repeated measures ANOVAs with practice time unit (epochs or sessions) as within-subject predictor, allowing random intercepts for the participants. In the first session, while responses became faster in general ($p < .01$), pre-error speeding ($p < .001$) and post-error slowing were attenuated ($p < .01$). All three temporal trends persisted during sessions 2–8 ($p < .001$; $p < .001$; $p < .01$). At the same time, repetition facilitation became attenuated ($p < .001$) and the effect of the prediction from the internal sequence model (the HCRP) was increased ($p < .001$). Compared with the last training session, in the first epoch of the interference session (session 9), participants slowed down ($p < .001$) and the predictions from the HCRP were less expressed in their responses ($p < .001$).

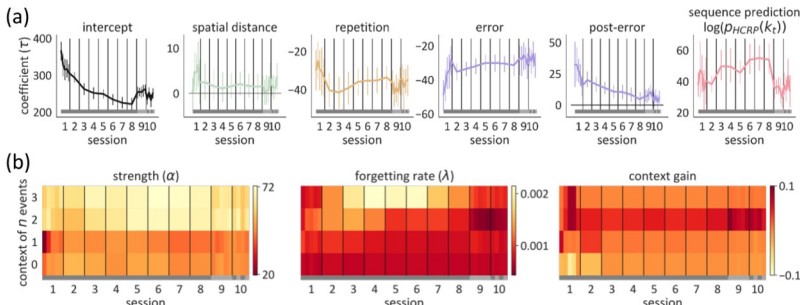

**Fig 4. Fitted parameter values shown session by session, and at a subsession resolution in the initial and final sessions.** The grey band on the bottom of each plot shows the sequence that participants practiced: the old sequence in sessions 1–8 (dark grey), the new sequence in session 9 (light grey), and both sequences alternately in session 10. Point distance in (a) and cell width in (b) are proportional to data bin size—we fitted the model to 5 epochs within sessions 1, 9, and 10 to assess potentially fast shifts. (a) Fitted values of the response parameters in units of $\tau$ [ms]. The error bars indicate the 95% CI for the between-subjects mean. (b) Fitted values of the strength $\alpha$ (left) and forgetting rate $\lambda$ (middle) parameters are shown, as well as their joint effect on prediction (right). A context of $n$ previous events corresponds to level $n$ in the HCRP. Lower values of $\alpha$ and $\lambda$ imply a greater contribution from the context to the prediction of behavior. The context gain for context length $n$ is the decrease in the KL divergence between the predictive distribution of the complete model and a partial model upon considering $n$ previous elements, compared to considering only $n − 1$ previous elements. Note that the scale of the context gain is reversed and higher values signify more gain.

## 2.2 Practice-related changes in the hyperparameters of the internal sequence model

The HCRP hyperparameters guide both learning and inference. Thus, the fitted HCRP hyperparameters reflect how participants use previous sequence knowledge as well as how they incorporate new evidence.

Remember that both $\alpha$ and $\lambda$ (Fig 4b, left and middle) determine the influence of a given context, by down-weighting shallower contexts or by reducing the decay of old observations in the same context, respectively. In order to visualize the joint effect of the two parameters, we computed the KL divergence between the predictive probability distribution given the whole context and shallower contexts. A lower KL divergence indicated that the context weighed strongly into the overall prediction. Then, we computed the degree to which the KL divergence is reduced by adding more context, and averaged it across all trials and contexts of a given length. The resulting values reflect the average *gain* from the context windows to the prediction of the response (Fig 4b, right).

During the first session, $\alpha_1$ increased ($p < .001$), suggesting that the first-order dependency of the response on the one previous event was reduced (Fig 4b, left, right). From session 2 to 8, both $\lambda_1$ and $\lambda_2$ decreased prominently ($p < .001$ and $p < .01$). This suggests that participants adaptively increased their memory capacity not just for first-order but also second-order sequence information. Storing more instances of their previous responses following two events allowed them to be more robust against local statistical fluctuations due to the random nature of the task. That is, their behavior became gradually less influenced by the most recent trigrams and better reflected the true trigram statistics. Even though third-order sequence knowledge (i.e. conditioning response on three previous events) would further improve performance, $\lambda_3$ did not decrease during training sessions 2 to 8 ($p = .376$.). This suggests that participants carved out the minimal context that is predictive and learned to remember previous instances in these contexts, ignoring deeper contexts that would give diminishing returns.

During session 9, when 75% of the sequence was changed, participants resisted the interference by further enhancing the trigram statistics that they had accumulated in earlier sessions, as reflected by further decrease in $\lambda_2$ ($p < .01$). With such a mild trigram forgetting, the internal sequence model remains dominated by data from the old sequence throughout sessions 9 and 10. At the same time, the sequence prediction coefficient was reduced sharply from session 8 to 9 ($p < .001$), indicating that the internal sequence model was not governing the responses as strongly anymore when it became largely outdated. This implies that the internal sequence model accounted for less variability in the RTs during the interference sessions than the late training sessions.

## 2.3 Correlation of the sequence model hyperparameters and working memory test scores

$\alpha$ and $\lambda$ capture how *sequence* memory is used. We conducted an exploratory analysis (Fig 5A and 5B) in order to relate the values of the HCRP hyperparameters that were inferred on the sequence learning task to participants' performance on independent memory tests. Three tests were conducted prior to the sequence learning experiment: the digit span test to measure verbal working memory (WM); the Corsi blocks test to measure visuo-spatial WM; and the counting span test to measure complex WM. The former two, 'simple' WM tasks require the storage of items, while the latter complex WM task requires storage *and* processing at the same time.

We found that the complex WM score was negatively correlated with $\alpha_3$ ($r = -.50$, $p = .01$) such that higher complex WM was related to more reliance on longer contexts for prediction

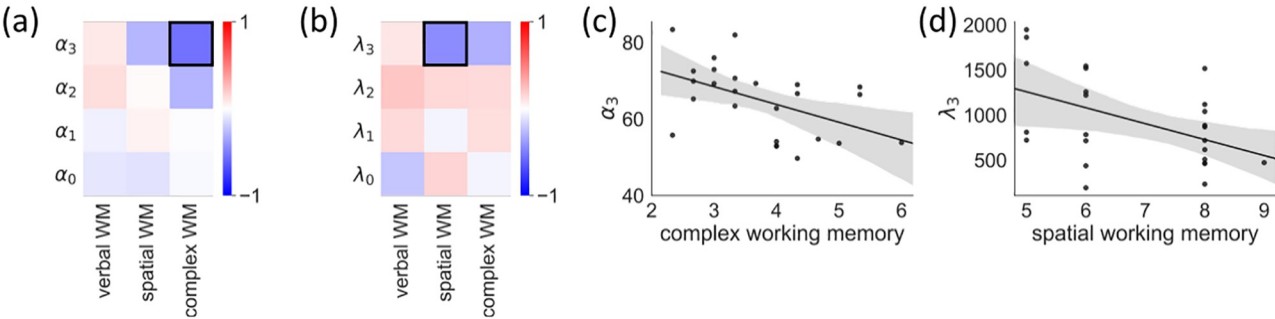

**Fig 5. Correlation between the fitted HCRP parameters and working memory.** (a)(b) Pearson correlation matrices of the working memory test scores and the strength parameters $\alpha$ and decay parameters $\lambda$ of the HCRP model, respectively. Correlations that met the significance criterion of $p <$ .05 are marked with black boxes. (c)(d) Scatter plots of the correlations that that met the significance criterion of $p <$ .05. Bands represent the 95% CI.

(Fig 5c). The spatial WM score was related to $\lambda_3$ ($r = -$ .47, $p =$ .02), reflecting that a higher spatial WM capacity allowed for better retention of previous responses in longer contexts (Fig 5d). Verbal WM was not related to any of the hyperparameters (all $p$s $>$ .05), probably due to the fact that the sequence learning task itself was in the visuo-spatial and not the verbal domain. In case we control for all 24 comparisons presented in the paper, the two significant correlations do not survive the Benjamini-Hochberg correction (their adjusted $p$ values increase to .09 and .28, respectively). However, we note that this correction is too harsh, as we did not have prior hypotheses about all of these 24 relationships. In fact, it was expected that the hyperparameters governing the role of very short contexts is not related to WM, as the 1–2 item WM capacity variance is expected to be extremely low in healthy adults. However, we included all comparisons for completeness.

## 2.4 Trial-by-trial prediction of response times

Using the fitted HCRP parameter values HCRP for each participant and session or epoch, we generated predicted RTs and evaluated the goodness of fit of our model by computing the coefficient of determination ($r^2$) between the predicted and measured RTs on held-out test segments. Fig 6a and 6b show how the predictions from the internal sequence model, as well as other, low-level effects jointly determine the predicted RTs on the first and the seventh training sessions of an example participant, respectively. In the first session (Fig 6a, Top), the predicted RTs (red line) were only determined by a slight effect of spatial distance between subsequent events and errors (pale green and purple bars). The internal sequence model was insufficiently mature to contribute to the responses yet. By session 7 (Fig 6b, Top), the sequence prediction effect (pale red bars) became the most prominent. Responses that previously were highly contingent on some part of the deepening event context were faster. This came from a well developed internal sequence model whose predictions became more certain and more aligned to the sequence of events (Fig 6b, Middle). By virtue of the HCRP, the depth of the substantially predictive context changed trial by trial (Fig 6a, bottom). On high probability trigram trials (marked by ticks) a context of two previous events was used for prediction, whereas on other trials only one previous event had substantial weight. Overall, this participant's responses in this late stage of learning became more influenced by the internal sequence predictions than the effects of spatial distance, error, and response repetition. On average across participants, the fraction of response time variance accounted for by the internal sequence prediction

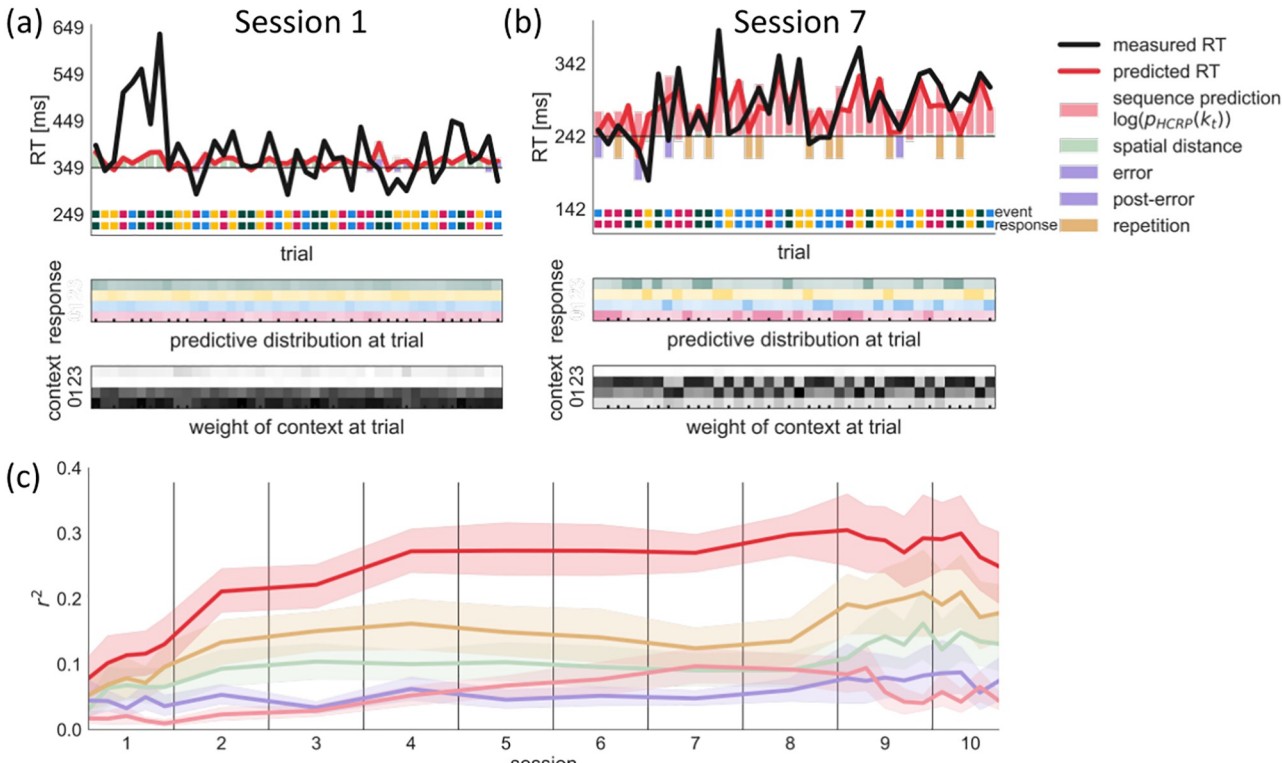

**Fig 6. Trial-by-trial predictive check.** In (a) and (b) we show example segments of held-out data from session 1 and 7 of participant 102 (Top) Colored bars show the positive (slowing) and negative (speeding) effects predicted by the different components in our model relative to the intercept (horizontal black line). The overall predicted RT value (red line) is the sum of all effects. The color code of the event and the response are shown on the bottom. A mismatch between the two indicates an error. (Middle) Predictive probabilities of the four responses are shown for each trial. The cells' hue indicate the response identity, saturation indicates probability value. The sequence prediction effect (pale red bar in (Top)) is inversely proportional to the probability of the response, i.e. higher probability yields faster response. The ticks at the bottom indicate high-probability trigram trials. (Bottom) We show what proportion of the predictive probability comes from each context length. Higher saturation indicates a larger weight for a context length. (c) Test prediction performance of the full model and each model component in terms of unique variance explained, averaged across participants. Bands represent the 95% CI.

increased monotonically from session 1 to 7; it plateaued by session 8 and reduced in the interference sessions 9 and 10 (Fig 6c).

## 2.5 The internal sequence model predicts second-order effects during learning and interference

Our model accounted for participants' sequence learning largely by enhancing the memory for trigrams of two predictive elements and a consequent response. This suggested that the HCRP, by virtue of its adaptive complexity, boiled down asymptotically to a stationary second-order sequence model (i.e. trigram model) with deterministic (*d*) and two sorts of random (*r*) trials, those following the deterministic scheme (random high; *rH*), and those not (random low; *rL*). Therefore, we tested how well calibrated the HCRP was to the second-order structure by analyzing the predicted RT differences on the held-out data, contingent on the sequence state and trigram probability. This follows the sort of descriptive analyses conventionally conducted in ASRT studies (e.g., [31]). We conducted two-way repeated measures ANOVAs with *time unit* (session or epoch) and *trial type* (state: *d/r* or P(trigram) in *r* states: *rH/rL*) as within-subject factors and with the measured or predicted RTs as outcome variable.

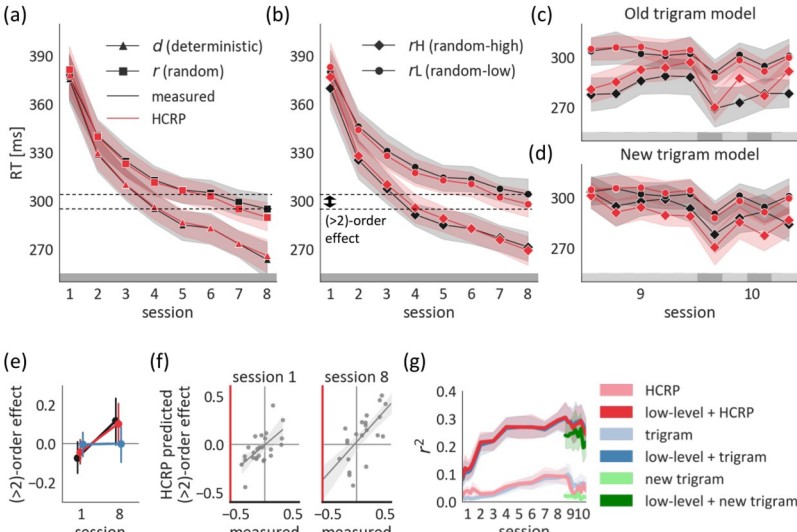

**Fig 7. Calibration of the HCRP model.** (a) RTs predicted by our HCRP model are shown against measured RTs for *d* versus *r* trials on held-out test data. (b) Same as (a) for *r*H versus *r*L trials. The two dashed lines mark the mean RTs for *d* and *r*H trials in session 8. The RT advantage of *d* over *r*H by session 8 marks ($> 2$)-order sequence learning. (c-d) *r*H versus *r*L trials are labelled according to the old trigram model (i.e. old sequence) or the new trigram model (i.e. new sequence). The grey band on the bottom shows the sequence that participants practiced: the old sequence in sessions 1–8 (dark grey), the new sequence in session 9 (light grey) and alternating the two sequences in session 10. (e) ($> 2$)-order sequence learning, quantified as the standardized RT difference between *r*H and *d* trials, shown for measured and predicted RTs. In session 1, *r*H trials are more expected because they reoccur sooner on average. By session 8, *d* trials are more expected because they are more predictable, given a $> 2$ context. This was predicted by the HCRP but not the trigram model. (f) Correlation of the measured and predicted ($> 2$)-order effect in session 1 and session 8. (g) Average predictive performance of the HCRP and the trigram models. (a-g) The error bands and bars represent the 95%CI.

During training sessions 1–8, participants gradually became faster for *d* than *r* trials, as well as for *r*H than *r*L trials. These divergence patterns were matched by the HCRP predictions ([Fig 7a and 7b](); significant *session*\**trial type* interactions in [Table 2]()). In the interference sessions 9 and 10, we tested the relationship between the RTs and the trigram probabilities for both old and new sequences. In order to study the effect of the old and new sequence statistics separately, we only included non-overlapping H trials (trials that are H in the old trigram model and L in the new one and vice versa) and contrasted them with overlapping L trials (trials that are L in both the old and new trigram models). In these analyses, we only consider *r* trials but we drop the *r* from the trial type notation for brevity.

In session 9, the effect of the old sequence statistics progressively waned, as evidenced by the coalescence of the curves for H$_\text{old}$ and L$_\text{old}$ trials ([Fig 7c](); H$_\text{old}$: diamond, L$_\text{old}$: circle). This

**Table 2. Repeated measures ANOVAs in sessions 1–8.** In the left set of columns, the *trial type* is defined as the state and in the right set of columns it is defined as P (trigram).

|  | effect | B [ms] | F | p | effect | B [ms] | F | p |
|---|---|---|---|---|---|---|---|---|
| measured RT | *session* | -10.11 | 133.38 | <.001 | *session* | -9.33 | 97.81 | <.001 |
|  | *state* | -1.99 | 116.01 | <.001 | *P(trigram)* | -13.68 | 272.66 | <.001 |
|  | *session\*state* | -3.63 | 29.07 | <.001 | *session\*P(trigram)* | -2.72 | 9.25 | <.001 |
| predicted RT | *session* | -11.06 | 216.68 | <.001 | *session* | -10.19 | 203.70 | <.001 |
|  | *state* | -3.16 | 171.44 | <.001 | *P(trigram)* | -7.02 | 229.83 | <.001 |
|  | *session\*state* | -2.76 | 44.92 | <.001 | *session\*P(trigram)* | -3.01 | 26.44 | <.001 |

**Table 3. Repeated measures ANOVAs in session 9 and 10.** In the left set of columns, the *trial type* is defined as $P_{old}(trigram)$ and in the right set of columns it is defined as $P_{new}(trigram)$.

| | effect | F | p | effect | F | p |
|---|---|---|---|---|---|---|
| session 9 | | | | | | |
| measured RT | *epoch* | .63 | .640 | *epoch* | .54 | .705 |
| | $P_{old}(trigram)$ | 91.48 | <.001 | $P_{new}(trigram)$ | 5.58 | .023 |
| | $epoch^*P_{old}(trigram)$ | 2.23 | .071 | $epoch^*P_{new}(trigram)$ | 1.02 | .396 |
| predicted RT | *epoch* | 3.65 | .008 | *epoch* | 3.17 | .016 |
| | $P_{old}(trigram)$ | 121.31 | <.001 | $P_{new}(trigram)$ | 98.38 | <.001 |
| | $epoch^*P_{old}(trigram)$ | 7.16 | <.001 | $epoch^*P_{new}(trigram)$ | 5.09 | <.001 |
| session 10 | | | | | | |
| measured RT | *epoch* | 2.29 | .085 | *epoch* | 2.50 | .036 |
| | $P_{old}(trigram)$ | 122.57 | <.001 | $P_{new}(trigram)$ | 21.71 | .002 |
| | $epoch^*P_{old}(trigram)$ | 1.49 | .223 | $epoch^*P_{new}(trigram)$ | 1.96 | .545 |
| predicted RT | *epoch* | 18.45 | <.001 | *epoch* | 15.00 | <.001 |
| | $P_{old}(trigram)$ | 207.71 | <.001 | $P_{new}(trigram)$ | 152.93 | <.001 |
| | $epoch^*P_{old}(trigram)$ | 5.13 | .003 | $epoch^*P_{new}(trigram)$ | 1.31 | .276 |

temporal pattern reflecting unlearning was significant for the RTs predicted by our model, but, being noisier, was only a trend for the measured RTs (Table 3 top left). The gradual divergence pattern of $H_{new}$ and $L_{new}$ typically seen in naïve participants was not significant for the measured RTs, contrary to the clear predicted relearning pattern predicted by our model (Fig 7d; Table 3 top right). Nevertheless, a slight overall speed advantage of $H_{new}$ over $L_{new}$ was also significant for the measured RTs, confirming overall learning in spite of the noisy learning curves. Indeed, by the last epoch of session 9, the mean speed advantage of $H_{new}$ to $L_{new}$ was not significantly lower than that of $H_{old}$ to $L_{old}$ (8.34 ms versus 13.91 ms, $t = 1.42$, $p = .166$), suggesting substantial learning but also resistance to interference. Surprisingly, the speed advantage of $H_{old}$ the in the last epoch of session 9 was positively correlated with $H_{new}$ ($r = .73$, $p < .001$). This suggests that the more efficient participants were at acquiring the old sequence, the better they learned the new one, suggesting a common factor behind the ability of parallel learning of new information and maintenance of old information. Our HCRP model could not account for this parallel process, because the forgetting mechanism inherently traded off the retention of old statistics against adapting to new statistics ($r = .16$, $p = .438$).

Due to the resistance to interference, the old trigram statistics were reactivated upon experiencing the old sequence. In the first epoch of session 10, participants' behavior was significantly influenced by the old sequence statistics, albeit to a lesser degree than prior to interference, in session 8 (mean RT difference between H and L was 32.87 ms and 21.43 ms, respectively; *time unit*$^*P_{old}(trigram)$ interaction: $p = .045$) and the amount of forgetting was closely estimated by our model (28.61 ms and 18.08 ms, respectively; $p = < .001$). Throughout the four alternating epochs of session 10, the measured RTs reflected the parallel maintenance of the two sequence representations, as the main effects of both $P_{old}(trigram)$ and $P_{new}(trigram)$ were significant (Table 3 bottom). The two trigram effects were not temporally modulated on the measured RTs—overall, there was no change in the influence of the old versus new sequence statistics due to the alternation between the old and new sequences. Whereas our model could account for the main trigram effects, it could not account for their joint temporal stability. Since the maintenance of the new trigram statistics was reflected in the measured RTs in epoch 2, our HCRP model assumed that this new knowledge traded off against knowledge

of the old sequence. Therefore, it incorrectly predicted weaker expression of the old sequence statistics in the epochs where the new sequence was practiced (notice the zig-zag pattern in the red lines in Fig 7c; *epoch*$^*P_{old}$*(trigram)* interaction in Table 3). The participants may have been able to employ meta-learning and invoke either the representation of the old or new sequence adaptively. Such a process could be captured by two independent HCRP sequence models and an arbiter that controls the influence of each, potentially based on their posterior probability. However, a perfect account of the resistance to interference and the parallel learning of two sequences is beyond the scope of the current paper.

## 2.6 The internal sequence model predicts ($> 2$)-order effects

Overall, the HCRP model captured the gradual emergence of second-order sequence knowledge and its resistance to interference. During sessions 1–8, it even captured higher-order effects, despite the fact that these are much weaker on average and more variable across participants. To assess this, we quantified a ($> 2$)-order effect as the (normalized) RT difference between *r*H and *d*, as is conventional in ASRT studies (e.g., [11, 32]). The reason is that *d* trials are constrained by the sequence phase, thereby respecting the ($> 2$)-order dependencies of the sequence whereas the *r*H trials are not constrained by the sequence phase and do not have ($> 2$)-order dependencies. Participants showed a reversal in the ($> 2$)-order effect whereby they were faster on *r*H than *d* trials in session 1 and they became faster on *d* than *r*H trials by session 8 (*session*$^*$*trial type* interaction: $p = .007$). This reversal could not be explained by a stationary trigram model that is, by design, agnostic to ($> 2$)-order dependencies and recency ($p = .939$; Fig 7e), but could be explained by the HCRP ($p = .025$).

The explanation based on the HCRP depends on the change in trigram forgetting. Since the *r*H trials are not locked to the sequence phase, their average reoccurrence distance is shorter than that of the *d* trials. In other words, if a trigram reoccurs in a *r*H trial, it tends to reoccur sooner than it would have in a *d* trial, at the appropriate sequence phase (S3 Fig). Therefore, stronger forgetting induced a slight speed advantage for *r*H trials in session 1, although this effect was not significant across the whole sample on either the measured RTs, or on those predicted by our model ($p = .084$ and $p = .199$, respectively). Individual differences in the initial recency bias were explained by the HCRP ($r = .620$, $p < .001$; Fig 7f, left). By session 8, trigram forgetting was reduced and the 4-gram statistics had a slight effect on behavior, as expressed by the advantage of *d* to rH trials. Due to heterogeneity among the participants, neither the measured nor the predicted average ($> 2$)-order effect was significant from zero on session 8 ($p = .080$ and $p = .076$, respectively). As in session 1, the individual variability in the ($> 2$)-order effect was captured by the HCRP ($r = .766$, $p < .001$; Fig 7f, right).

Nevertheless, across all sessions, the ($> 2$)-order effect was rather small compared to the second-order effect, even though learning the ($> 2$)-order dependencies allowed for more certain predictions. This can be viewed as resource-rational regularisation of participants' internal sequence model. Therefore, overall, the HCRP approximated a stationary trigram model and these two models explained a similar total amount of variance in the RTs (Fig 7g).

## 2.7 Predicting the response time of errors

So far, we accounted for general pre-error speeding and post-error slowing in the linear response model. By doing so, we controlled for factors other than sequence prediction influencing the error latency, for instance, a transiently lower evidence threshold (see e.g., [33]). Sequence prediction influences the latency of errors by providing what amounts to prior evidence for anticipated responses. As such, errors reflecting sequence prediction are predicted to be even faster than errors arising from other sources (e.g., within-trial noise of evidence

accumulation rate, not modeled here). To assess this, we categorised participants' errors based on the sequence prediction they reflected. 'Pattern errors' are responses that are consistent with the second-order dependencies, i.e. the global statistics of the task, instead of the actual stimulus. For instance, the following scenario was labelled as a pattern error: 'red'-X-'blue' was a second-order pair in the task and on a random trial that was preceded by 'red' two time steps before the participant incorrectly responded 'blue' when the event was 'yellow'. 'Recency errors' are repetitions of the most recent response given in the same context of two previous elements, i.e. they reflect sensitivity to local trigram statistics. Only responses that did not qualify as pattern error could be labelled as recency errors. For instance, the following scenario was labelled as a recency error: on the most recent occurrence of the context 'red'-'red', the participant responded 'yellow'; in the current context, the participant incorrectly responded 'yellow' when the event was 'green'. Errors that fell into none of these two categories were labelled as 'other'. We only analyzed those errors that were made on low-probability trigram trials. The proportion of pattern errors gradually increased due to learning, while the proportions of recency errors and other errors was reduced (Fig 8).

Paired $t$-tests revealed that the measured RTs were faster for pattern errors than other errors ($RT_{other} - RT_{pattern}$ = 27.55 ms, $t$ = 8.58, $p < .001$), suggesting that expectations based on the global trigram statistics indeed contributed to making errors. Similarly, participants were 12.84 ms faster to commit recency errors than other errors ($RT_{other} - RT_{recency}$ = 11.47 ms, $t$ = 2.38, $p$ = .025) (Fig 9). This indicated that participants' errors were also influenced by local statistics, although to a significantly lesser degree ($t$ = 2.76, $p$ = .011). While a stationary trigram model was able to explain the pattern error RTs ($RT_{other} - RT_{pattern}$ = 25.15 ms, $t$ = 11.04, $p < .001$), it lacked the distance-dependent inference mechanism that could explain the recency error RTs ($RT_{other} - RT_{recency}$ = -1.14 ms, $t$ = -1.18, $p$ = .249).

The HCRP model correctly predicted fast pattern errors ($RT_{other} - RT_{pattern}$ = 21.32 ms, $t$ = 20.91, $p < .001$), but underestimated the speed of recency errors ($RT_{other} - RT_{pattern}$ = 4.67 ms, $t$ = 3.91, $p < .001$, Fig 9). The reason for this was that the HCRP fit the data by reducing trigram forgetting, explaining participants' overall robustness to the recent history of trigrams

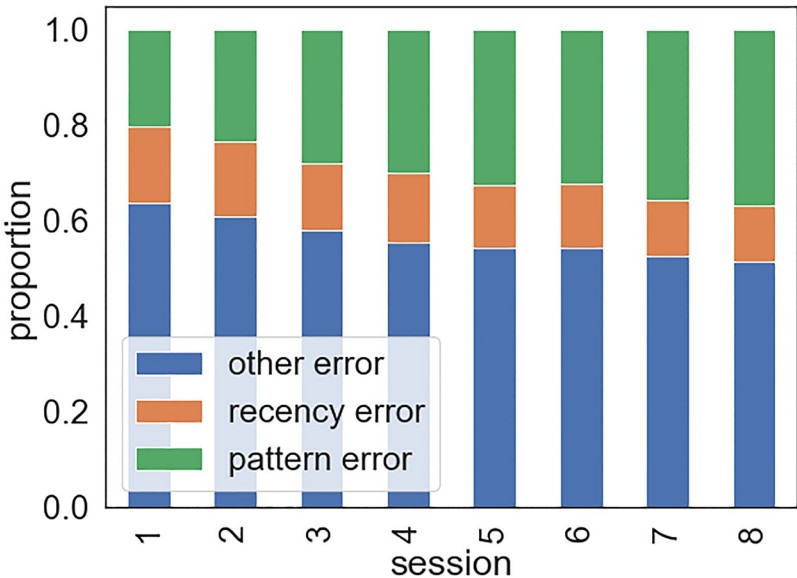

**Fig 8. Proportions of errors of different types across training sessions.**

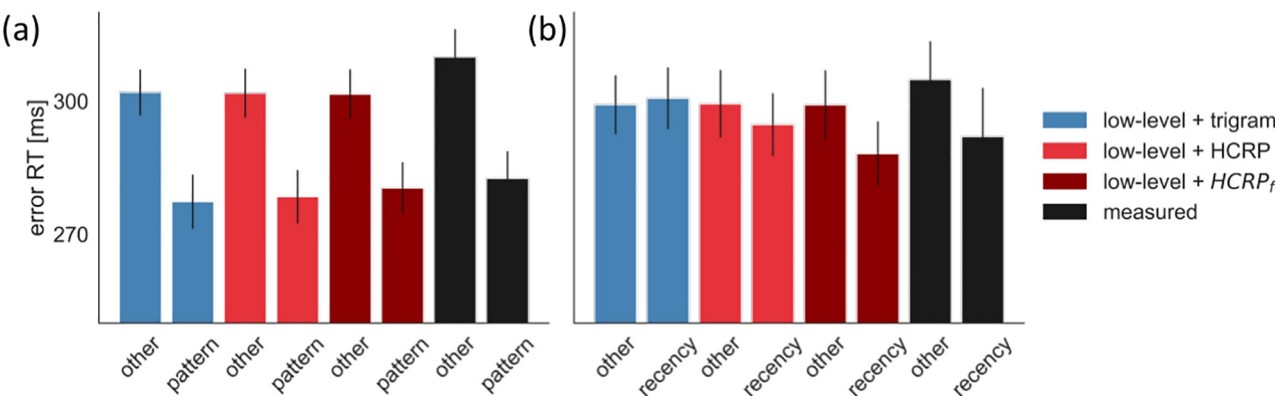

**Fig 9. Predicting the latency of errors.** (a) Pattern errors. (b) Recency errors. In the case of HCRP$_f$, the hyperparameter priors were adjusted to express more forgetfulness. The error bars represent the 95%CI.

(Fig 4). However, as our error RT analysis revealed, sensitivity to recent history was more expressed on error trials.

Explaining why error responses were more influenced by recent history than correct responses is beyond the scope of this paper. However, note that our HCRP model does have the flexibility to explain the effect in isolation. Therefore, we fitted the HCRP to the same data but using a different prior for the forgetting hyperparameters, thus projecting the model into a more forgetful regime (priors defined in S1 Table; posteriors shown in S4 Fig). As shown in Fig 9, the HCRP with stronger forgetting prior, HCRP$_f$, was able to explain the degree to which error responses were influenced by global and local trigram statistics, as the speed advantage of pattern errors ($RT_{other} - RT_{pattern} = 21.79$ ms, $t = 11.10$, $p < .001$) and recency errors ($RT_{other} - RT_{recency} = 10.92$ ms, $t = 6.33$, $p < .001$) was more correctly predicted on average. In sum, while a less forgetful higher-order sequence learning model accounted best for participants' RTs due to their general robustness to local noise, a more forgetful model accounted for a slight sensitivity to local noise that was expressed in the speed of error responses.

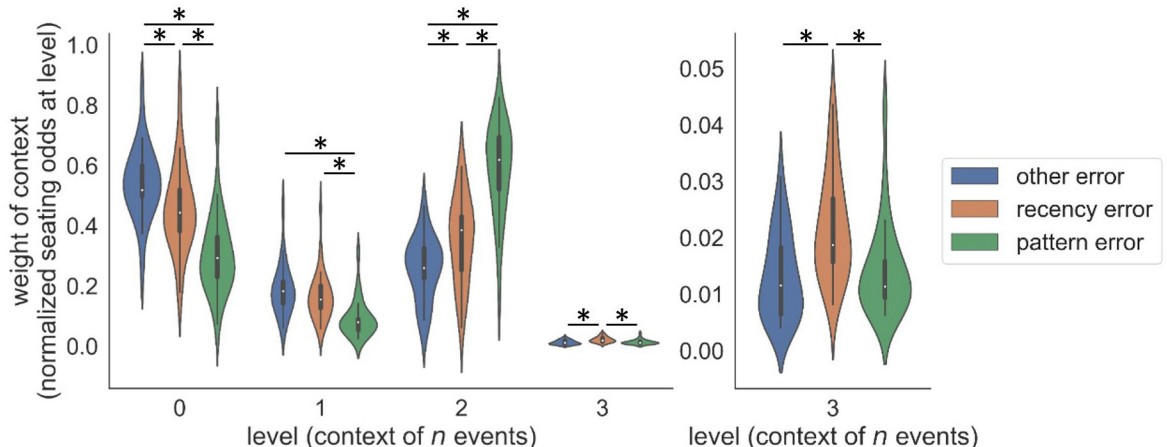

**Fig 10. The HCRP$_f$ weighting of context depth is different among errors of different types.** (Left) Weights of all HCRP$_f$ levels. (Right) Zoomed in for HCRP$_f$ level 3 only.

In order to elucidate the relationship between the inferred seating arrangements in the HCRP$_f$ model and participants' errors, we computed the average seating odds for each level across the three error types (the same measure as in Fig 6a and 6b, bottom). As shown in Fig 10, the weight of level 2 in the HCRP$_f$, that is, the recency of trigrams, did not influence the speed of different types of errors to the same degree ($F(2, 72) = 39.90$, $p < .001$). The weight of level 2 was stronger in the case of recency errors than other errors ($t = -2.72$, $p = .009$), and it was even stronger in pattern errors than recency errors ($t = -5.61$, $p < .001$). The overall difference in the weight of level 0, that is, the recency of unigrams ($F(2, 72) = 20.94$, $p < .001$) was driven by the opposite trend. The weight of the unigram observations was stronger when participants committed other errors than recency errors ($t = 2.35$, $p = .02$), and stronger in the case of recency errors than pattern errors ($t = 3.89$, $p < .001$). There were significant differences among the error types in the weights of level 1 and 3 as well, though more modest and not three-way ($F(2, 72) = 10.26$, $p < .001$; $F(2, 72) = 8.02$, $p < .001$, respectively). These results demonstrate the straightforward relationship between the learned parameters of the HCRP$_f$, i.e. proportions of observations contingent on deepening contexts, and the types of errors that participants made.

## 2.8 Alternative models

The principled back-off mechanism of the HCRP model was essential to account for participants' smoothing behavior—namely, that they flexibly combined higher- and lower-order information for prediction. Ablated alternative models listed in A Table in S2 Appendix, even if containing higher-order sequence information, fell short in explaining how participants combined the higher-order information locally, while also maintaining stable knowledge of the global chunk statistics (D Fig, E Fig and F Fig in S2 Appendix).

## 3 Discussion

In this study, we elucidated how humans come to exploit intricate regularities in their sensori-motor environments. We used a hierarchical non-parametric Bayesian model to characterize the gradual, implicit learning of the higher-order sequential structure in a serial reaction time task over thousands of trials. Our model fitted the trajectory of response times, showing how participants refined their internal sequence models, adapting to larger and larger predictive chunks. Model parameters correlated with working memory capacity and error characteristics.

As a generative model for sequences, the hierarchical Bayesian sequence model [14] that we used is able to capture all these dependencies. More than that, it combines the multi-order information in an adaptive way. As a recognition model for accounting for the reaction times of our participants, judicious choices of priors per session, adjustment of the model to allow for forgetting [23], and augmentation with low level effects such as for repetitions and error, allowed it to fit behaviour rather well (and better than alternative models lacking an adaptive chunk-smoothing mechanism). Updating the model after every observation enabled it to track learning at a trial-by-trial resolution. The hyperparameters of the model that govern the preference to use longer contexts and to be less forgetful were correlated with participants' complex and spatial working memory scores, as measured by independent tests. [34] suggested that there is no substantial influence of working memory capacity on sequence learning if the task is implicit. However, their review included studies in which the analyses were based on aggregate sequence learning scores (computed as the response time difference between predictable and unpredictable trials). Here we show that the parameters of a mechanistically modeled sequence forgetting process are in fact related to working memory. This highlights that working memory does play a role in sequence processing—

whether explicit or implicit—although a relatively elaborate modeling might be needed to capture its subtle and/or complex contributions.

According to the HCRP, participants gradually enriched their internal sequence model so that it reflected zero and second-order contingencies. However, in the first session, forgetfulness-induced volatility in the sequence model explained why high probability trigram trials were more expected by the participants in random states. In previous ASRT studies, this effect was termed 'inverse learning' [17, 35]. Here we show that the effect is not counter-intuitive if we allow for forgetfulness. The effect arises due to the specific distance structure of the ASRT, namely that the same trigram recurs at smaller distances, on average, in random trials than deterministic trials (that enforce spacing among the elements; S3 Fig). As such, the 'random high' trials were more readily recalled.

As sessions progressed, trigram forgetfulness was reduced. This can be seen as a form of long-term skill learning as more importance and less forgetting is gradually assigned to predictive context-response chunks. Thus, after the first session, participants did not expect a globally frequent trigram less if that trigram happened to be locally rare. Previous work highlighted one key aspect of well-established skills: that the variability of self-generated actions is decreased, yielding smoother and more stable performance [36]. By contrast, here, skill is associated with more sophisticated methods of managing regular external variability. Simply put, learners can change the size of chunks they choose to remember better. The context-depth specific parametrisation of our model allows for the identification of other potential practice-related changes, which were actually not observed in the ASRT data, but are often observed in other tasks. For instance, in case of higher memory demands, it is possible that participants improve their performance by shifting the strength of a chunk size required by the task but remain forgetful about chunks of that size. This would result in behavior that is increasingly contingent on the correct chunk size but noisy due to over-reliance on recent observations.

By the last training session, some participants enriched their sequence representation further, with increased memory not only for trigrams but also for four-grams, implicitly incorporating knowledge of the sequence phase and enabling better performance. This shift was much milder than that in the memory for trigrams, and was not exhibited by all participants. While many previous studies have focused on the question of whether higher-order learning did, or did not, take place under certain conditions, special groups, etc., our method uncovers the *degree* to which information of different orders was learned.

Our model was able to account for the initial resistance to interference and slow relearning when a new sequence was introduced in session 9. Sequence interference not only reduced the response time variance explained by the internal sequence model (which was full of information about the old sequence), but also increased the variance explained by low-level effects, such as the response repetitions and the spatial distance between the current and previous cue. However, note that the low-level *effect sizes* did not increase—in fact, they mildly decreased as a consequence of the interference, as shown in Fig 4a. Therefore, our interpretation is that participants did not 'fall back' to rely on aspects of the data other than chunk statistics (in which case our labeling of these effects as 'low-level' might be brought into question), but rather that the low-level effects were less obscured by the effects of learning that became obsolete during interference.

Finally, our model class could reproduce error speeding that was specific to the type of errors, that is, whether they reflected the global statistics, local statistics, or no apparent statistics of the task. However, since there were insufficient errors (~10%) in our data for them to exert sufficient impact on the parameters, to examine a ddHCRP account of go wrong more precisely, we had to force the model into a more forgetful regime by adjusting the minimum forgetting rate to a lower level in the prior. Whilst acknowledging the artificiality of this

procedure, we hope that the result is relevant for cases such as explicit sequence prediction tasks in which erroneous responses are more prevalent.

The HCRP model has distinct advantages over the two alternatives that have been suggested for characterizing or modeling the ASRT. Until recently, all papers employing the ASRT stuck to the kind of descriptive analysis that was used in the first ASRT paper [11]. This purely descriptive account focuses rather narrowly on the task itself, asking whether participants' behaviour is appropriately contingent on the frequent and infrequent trigrams that arise from the structure of the task. Although such descriptions can show what sort of conformance there is, and how quickly it arises over sessions, they do not provide a trial-by-trial account of learning. In these terms, trigram dependencies arise more strongly in the HCRP as forgetfulness wanes—something that happens relatively quickly across sessions, underpinning the general success of the trigram model in explaining behaviour.

The other, contemporaneous alternative account [15] is mechanistic, and so is rather closer to ours. [15] fitted their model to the same data set that we present here, therefore the differences between our models and fitting procedures are straightforwardly understood and we will elaborate on them here. They use an infinite hidden Markov model (iHMM) [37, 38] which is a nonparametric Bayesian extension of the classical HMM, capable of flexibly increasing the number of states given sufficient evidence in the data. As such, it can automatically 'morph' into the generative process of the ASRT, that is, into a model containing four states, each deterministically emitting one event, deterministically alternating with four random states, each emitting either of four events with equal probability. The trouble with the iHMM for modeling variable length contextual dependencies is that it has to use a proliferation of states to do so (typically combinatorial in the length of the context), posing some of the same severe statistical and computational difficulties as $n$-gram models that we noted in the introduction. Indeed, unlike the case for HMMs, parts of the sequence that have not proved predictive or have proved superfluous—whether beyond or even intervening in the predictive window—are organically under-weighed in the HCRP. As another alternative, a sequence compression method was proposed for modeling compact internal representations [39], however, this method is yet to be extended to non-binary sequences and trial-by-trial dynamic compression. Our model, apart from capturing parsimony, also captured the nonstationarity of humans' expectations. Thus, it explained higher-order perseveration effects whereby recent chunks were more expected to reoccur.

Apart from the prior structure itself, our modeling approach differed from that employed by [15] in three ways. First, [15] assumed a stationary model structure for each session, while we sought to capture how participants update their internal model trial by trial. This is important for the initial training session and the interference sessions, where quick within-session model updating is expected. Second, instead of treating the learning sessions independently, we assumed that participants refine the same internal model session by session with new information, as well as by adjusting the hyperparameter set of that model (i.e. *how* the learned information used). Third, we controlled for the rather strong low-level effects that are ubiquitous in sequence learning studies and are often controlled for in descriptive analyses: the effect of spatial distance of response cues, repetition facilitation, pre-error speeding, and post-error slowing. Although some of these can, in principle, be characterized by the HCRP and the iHMM, it is likely that they are generated by rather different mechanisms in the brain (e.g., repetition effects may arise from the modulation of tuning curves [40]), and so it can be confounding to corrupt the underlying probabilistic model learning with them. Of course, separating low-from high-level effects is not always so straightforward.

Serial reaction time tasks of various sorts have been extensively used to assess whether sequence learning can be implicit [1]; developmentally invariant or even superior in children

compared to adults [41]; impaired [42], intact [43, 44] or even enhanced [45] in neurological and psychiatric conditions; persistent for months [31] and resistant to distraction [12] even after brief learning; (in)dependent on sleep [32, 46] and subjective sleep quality [47], etc. It would be possible to use the parameterization afforded by models such as the HCRP to ask what components differ significantly between conditions or populations.

We treat the ddHCRP as a computational-level description of multi-order sequence statistical learning and use, rather than as a process model that could be transparently implementable in neural hardware. Non-parametric Bayesian methods in this family are quite prevalent as computational-level models in cognitive science (e.g., the work of [48] on extinction in classical conditioning; and [49] in motor control), mostly also without suggested neural implementations. However, there has been at least one interesting attempt to link Chinese restaurant processes to cortico-striatal interactions by [50]. They developed a cognitive model of structure learning based on the CRP, akin to our model, along with a neurobiologically explicit network model approximation. Later, they demonstrated that EEG signals were predictive of participants' CRP-like clustering behavior [51]; along with fMRI-based investigations about prefrontal cortical regions involved in cluster creation and use [52]. One could perhaps imagine that the rich complexities of the expansive and contractive connections between the cortex, the basal ganglia and the striatum could implement some form of the hierarchy in the ddCRP. Alternatively, purely cortical mechanisms might be involved.

Given the utility of the HCRP model for capturing higher-order sequence learning in humans, it becomes compelling to use it to characterize sequential behavior in other animals too. For instance, the spectacular 'dance show' of a bird-of-paradise, the western parotia, comprising a series of different ballet-like dances has been recorded extensively but is yet to be modeled. There has been more computational work on bird songs. Bengalese finches, birds that are domesticated from wild finches, developed probabilistic, complex songs that a first-order Markov model of the syllables is not able to capture [53]. [54] modeled second-order structure of the Bengalese finch songs using a partially observable Markov model where states are mapped to syllable pairs. However, some individuals or species might use even higher-order internal models to generate songs, and this would be a natural target for the HCRP.

Structures in birds, such as area HVC, that apparently sequence song [55], or the hippocampus of rodents with its rich forms of on-line and off-line activation of neurons in behaviourally-relevant sequences [56, 57], or the temporal pole and posterior orbitofrontal cortex of humans [58], whose activity apparently increases with the depth of the predictive context, are all attractive targets for model-dependent investigations using the HCRP.

Of course, some behaviour that appears sequentially rich like that of the sphex, the jeweled cockroach wasps, the fruit flies [59] or indeed of rodents [60] in their grooming behaviour may actually be rather more dynamically straightforward, proceeding in a simple mandatory unidirectional order that does not require the complexities of the HCRP to be either generated or recognized. Thus, fruit flies clean their eyes, antennae, head, abdomen, wings and thorax, in this particular order. [59] showed that this sequence arises from the suppression hierarchy among the self-cleaning steps: wing-cleaning suppresses thorax-cleaning, abdomen-cleaning suppresses both of these, and head-cleaning suppresses all the others. In a dust-covered fly, all of the cleaning steps are triggered at the same time. But each step can only be executed after the neural population underlying it is released from suppression, upon completion of the cleaning step higher in the hierarchy. As such, the suppression hierarchy ensures a strictly sequential behavior. Notably, behaviour of this sort is closed loop—responding to sensory input about success—whereas we consigned closed loop aspects to the 'low level' of effects such as post-error slowing. It would be possible to tweak the model to accommodate sensory feedback more directly.

In conclusion, we offered a quantitative account of the long-run acquisition of a fluent perceptuo-motor skill, using the HCRP as a flexible and powerful model for characterising learning and performance associated with sequences which enjoy contextual dependencies of various lengths. We showed a range of insights that this model offered for the progressive performance of human subjects on a moderately complex alternating serial reaction time task. We explained various previously confusing aspects of learning, and showed that there is indeed a relationship between performance on this task and working memory capacity, when looked at through the HCRP lens. The model has many further potential applications in cognitive science and behavioural neuroscience.

## Supporting information

**S1 Appendix. Supplementary algorithms.**
(PDF)

**S2 Appendix. Supplementary results.**
(PDF)

**S1 Fig. Sequence predictions of 3-level HCRP models fitted to 100 data points.** The models were trained with batch learning in order to clearly show how the pattern of predictions depends on the sequence structure without online updates of the model parameters. In (a), the sequence was the concatenation of repeats of a 12-element determinstic pattern (Serial Reaction Time Task or SRT). In (b), the sequence was generated from the ASRT. (Top) Colors denote the sequence elements. The vertical bar marks the boundary between the two repeats in the SRT example segment. (Middle) Predictive probabilities of the four events are shown for each trial. The cells' hue indicate the event identity, saturation indicates probability value. The Xs indicate the event with the highest predicted probability, i.e. the predicted event; Xs are green for correct predictions and red for incorrect predictions. The ticks at the bottom in (b) indicate high-probability trigram trials. Note that, after having a context of at least two previous elements, all predictions are correct in the case of the deterministic SRT. In the ASRT, incorrect predictions occur for the low probability trigrams. (Bottom) We show what proportion of the predictive probability comes from each context length. Higher saturation indicates a larger weight for a context length. Note that the context of two previous elements is invariably dominant in the SRT predictions where every event is predictable from the previous two. In the ASRT, the context weights follow the largely alternating pattern of the high and low probability trigrams, the former ones being predictable from two previous events, the latter ones being unpredictable.
(TIFF)

**S2 Fig. Negative log likelihood loss of HCRP models fitted to 10.000 ASRT data points.** (a) Negative log likelihood as a function of the maximum number of previous events considered. (b) Negative log likelihood as a function of the prior importance of two previous events, i.e. trigrams (b). In (b), lower values of $\alpha_2$ imply higher prior importance. The vertical dashed line in (a) marks the $n$ that was used for fitting the human data in the Manuscript.
(TIFF)

**S3 Fig. Trigram reoccurrence distance in trials.** Vertical lines mark the medians. Note the marked periodicity in the case of $d$ trials that imposes a spacing among the trigrams and increases the median reoccurrence distance.
(TIFF)

**S4 Fig. Fitted values of the strength $\alpha$ (left) and forgetting rate $\lambda$ (middle) parameters, as well as their joint effect on prediction (right), using the constrained prior that places the model in a forgetful regime, described in S1 Table.** A context of n previous events corresponds to level n in the HCRP. Lower values of $\alpha$ and $\lambda$ imply a greater contribution from the context to the prediction of behavior. The context gain for context length n is the decrease in the KL divergence between the predictive distribution of the complete model and a partial model upon considering n previous elements, compared to considering only $n$-1 previous elements. Note that the scale of the context gain is reversed and higher values signify more gain.
(TIFF)

**S1 Table. Hyperparameter prior sets for fitting the response times of all responses (sections 3.2-.3.6) and errors only (section 3.7).** In session 1, the prior was uninformed. In all subsequent sessions, the prior was a truncated Gaussian $\mathcal{N}^{\prime}$ with the mean of MAP value in the previous session, a fixed variance, and the same interval that the uninformed distributions have in session 1. For most of our results, the first, wider of $\lambda$ prior was used to allow for extreme forgetfulness or unforgetfulness. For the prediction of response times of errors, we restricted our model to a more forgetful regime by narrowing the $\lambda$ prior.
(PDF)

**S2 Table. Mixed effects model with random intercepts for participants and several low-level predictors, sorted by their absolute fitted slope B (in ms).** Due to the large data set, all factors are significant. However, we made an arbitrary cut-off at the horizontal line for the low-level effects included in the response model because of the small effect sizes.
(PDF)

## Acknowledgments

We thank Sebastian Bruijns for helpful discussions on our model and Eric Schulz for providing useful feedback on the manuscript.

## Author Contributions

**Conceptualization:** Noémi Éltető, Dezső Nemeth, Karolina Janacsek, Peter Dayan.

**Data curation:** Noémi Éltető.

**Formal analysis:** Noémi Éltető.

**Funding acquisition:** Peter Dayan.

**Investigation:** Noémi Éltető, Dezső Nemeth, Karolina Janacsek.

**Methodology:** Noémi Éltető, Peter Dayan.

**Software:** Noémi Éltető.

**Supervision:** Peter Dayan.

**Validation:** Noémi Éltető, Peter Dayan.

**Visualization:** Noémi Éltető.

**Writing – original draft:** Noémi Éltető, Peter Dayan.

**Writing – review & editing:** Noémi Éltető, Dezső Nemeth, Karolina Janacsek, Peter Dayan.

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
