## [Decision Letter · Decision Letter 0]

20 Apr 2022

Dear Ms Éltető,

Thank you very much for submitting your manuscript "Tracking human skill learning with a hierarchical Bayesian sequence model" for consideration at PLOS Computational Biology.

As with all papers reviewed by the journal, your manuscript was reviewed by members of the editorial board and by several independent reviewers. In light of the reviews (below this email), we would like to invite the resubmission of a significantly-revised version that takes into account the reviewers' comments.

We cannot make any decision about publication until we have seen the revised manuscript and your response to the reviewers' comments. Your revised manuscript is also likely to be sent to reviewers for further evaluation.

Sincerely,

Christoph Mathys

Associate Editor

PLOS Computational Biology

Samuel Gershman

Deputy Editor

PLOS Computational Biology

Reviewer's Responses to Questions

**Comments to the Authors:**

Reviewer #1: This article proposes a novel computational model of sequence learning, and tests some predictions of the model using behavioural data acquired in a variant of the Alternating Serial Response Time (ASRT) task. On each trial, one out of four possible cues is shown and subjects (n=25) have to respond as quickly as possible by pressing a (cue-specific) key. Unbeknownst to the participants, half the cues follow a deterministic second-order sequence, such that the cue at trial t is exactly predictable from the cue at trial t-2. The other half (interleaved with the predictable cues) is random. Participants completed 8 weekly sessions of 2125 trials each (!) where the underlying deterministic sequence was kept fixed, plus 2 additional sessions that include covert changes in the deterministic sequence. Peoples' implicit learning in the task was monitored using response time, which essentially increase when they are surprised. The main result is twofold: (i) after learning, peoples' RT is quicker for predictable cues than for random cues, and (ii) they are also quicker for those random cues that conform (by chance) with the deterministic 2nd-order sequence than for the other random cues. Taken together, these results imply that participants have learned (some aspect of) the task contingencies, albeit entirely implicitly.

Overall, I found the paper interesting and timely. The modelling aspect of the work is novel and promising, and the methods consists of a balanced mixture of computational and experimental approaches, which I feel empathetic with. Having said this, there are a number of issues that, to my opinion, slightly compromise the quality of the paper (see below). I believe that, if these are addressed in the revised manuscript, this paper would make a significant contribution to the field. Let me now expose the two main concerns I have with the current version of the manuscript.

First of all, I felt that the experimental paradigm was not perfectly appropriate for testing the model. This is essentially because the model is about the ability to flexibility adapt the sophistication of sequence learning to the (hidden) complexity of the sequence. In brief, the model starts with the premise the brain should be equipped with some mechanism that enables it to gradually change its representational dynamics, sic : "We hypothesized that humans build their internal sequence representation in a similar way, starting with learning short-range dependencies and gradually adding long-range dependencies if they are indeed significantly present in the data" (l. 164-166, p. 6). Accordingly, the model focuses on an optimal (Bayesian) mechanism that can learn any dependency structure (this is the computational problem that the model is meant to solve). However, the dependency structure of the ASRT task is invariant. In other terms, one can solve the task without being able to flexibly adapt to the dependency structure. I would even argue that the main results would be expected, under any form of sequence learning that would capture 2nd-order dependencies. I note that this most likely extends to most (if not all) results reported in the manuscript as it stands. For example, a simple online regression based upon, e.g., truncated Volterra series would make qualitatively identical predictions. Now, I'm not saying people are doing this. Rather, I'm challenging the implicit assumption that the current experimental design offers direct empirical evidence for the model that authors have proposed here. In my opinion, this implies that authors should provide comparative evidence for their model. In other terms, they should try other (simpler?) models, and show that they are less likely explanations for peoples' behavior than the "distance-dependent Hierarchical Chinese Restaurant Process" (hereafter: ddHCRP) model. Importantly, the comparison should be fair, in that candidate models should be a priori able to learn the 2nd-order sequence structure.

Second, I have a few issues with the presentation of the model, as it stands. In brief, if one is not already cognisant of Dirichlet processes, then one cannot understand how the model works. There are only 3 equations in the manuscript that relate to the model, and they clearly are not sufficient to describe the model. This is PLoS Computational Biology: authors should not be reluctant to be explicit about the mathematics :) As I am sure authors are well aware, the typical way of describing a bayesian learning/inference algorithm is to start with the generative model, and then describe the model inversion procedure. Equations 1-3 are summarizing the main aspect of the generative model, but describing the full generative model requires more details. More importantly, no computational detail is given regarding model inversion. Does it rely on sampling, or some variational approximation (the latter, I guess, since they cite Beal and colleagues)? The authors should insert a complete model/methods subsection on model inversion, with full details regarding the algorithmic approach. For example: how is it initialized? What summary statistics are used and how are they updated online? How does computational complexity grow (Dirichlet processes rely on some threshold to augment their state-space: what is it here?) ? They should use this description to highlight the core computational properties of the ddHCRP learning algorithm. In relation to the first comment above, this may also serve to motivate the choice of candidate alternative models, and discuss possible pros and cons from a computational perspective.

I have other more minor concerns, but I don't believe it is useful to discuss these unless authors are willing to address those two issues first. I hope authors understand my concerns, and take this review round as an opportunity to improve the paper (which I think deserves to be published in PLoS CB, provided the above concerns are adequately addressed)!

Jean Daunizeau.

Reviewer #2: Learning increasingly complex statistical regularities present in

continuous sequences is a difficult problem affected by the curse of dimensionality. Yet, humans can take advantage of such regularities to speed up their responses during perceptuo-motor tasks. In this manuscript, the authors adapt an existing model of language processing to capture the learning process of participants practicing a visuo-motor task over multiple weeks.

Overall, the model elegantly capture many behavioural hallmarks of implicit sequence learning, and thus appear as a promising quantifying tool. The manuscript focuses on validating the predictions of the model, laying the groundwork for future studies in cognitive neurosciences.

I have two major comments and few minor points I would like the authors to address before considering publication.

Major comments:

- I might have missed some supplementary material, but I could not find the formal description of the model. I understand the hierarchical Dirichlet process is relatively standard, but the details of its implementation and in particular all the adaptations of the response mapping (low level biases), the explicit rules for selecting and weighting levels, and the inversion routine must be included, albeit in an annex, both for the sake clarity and to ensure self-sufficiency of the manuscript and therefore reproducibility. The absence of a complete set of equations describing the learning rules and the response function, combined with the lack of typographic distinction between scalar and vector parameters, makes the read rather difficult. The author should also make sure to make their code available in a public repository for completeness.

- The model is entirely fitted to the RTs of actual responses, which is perfectly understandable and well justified in the manuscript. However, the structure of the model should allow to also predict the "motor choices", and more interestingly, make non-trivial predictions about errors (eg. generalisation errors). The authors should provide some hints about this could be implemented or, better, provide some additional analyses addressing this point.

Having a look at such qualitative predictions might be particularly critical eg. in section 2.7 in which 'pattern errors' (coming from learned expectations) are opposed to 'recency' and 'other' errors: if the differential in RT between those cases is indeed coming from the expression of some learned >2-order contingencies, choices qualifying as 'pattern errors' should be aligned with the internal 'seating pattern' recovered by the learning model (but not in the case of other errors). Addressing this point would make a fair sanity check of the interpretation of the error speeding in pattern errors.

Another prediction would be that participants/phases characterised by a deeper representation should also exhibit specific types of error reflecting their higher order expectations. I understand such events might be rare and therefore hard to analyse, but they could provide some further insights into the behavioural variability across participants, which is a bit lacking in the current manuscript, especially for a model intended as a quantification tool for behavioural neurosciences.

Minor comments:

- I understand where the Chinese restaurant example comes from, but the back and forth between this terminology and the presented experimental design is a bit hard to follow. Sentences like 'the customers respond to certain key presses' or 'the probability of sitting at a table ... predict how likely each key is' are a bit nonsensical for a reader not familiar with Chinese restaurant processes. If the author insist on keeping the CRP example, I would suggest doing it in one place, then map each terms to both the equations (cf my first point) and the respective experimental concepts (eg. what is a table in terms of sequence representation?).

- p.10: "We parsed the sequence five times..." Why five? This seems a very low for a sampling procedure. Or is this meant at the trial level, therefore exhausting the possible seating arrangements?

- On the same topic, it would be helpful to have some measure of convergence of the inversion procedure (eg. variance of the estimates across different runs of the random search)?

- Is there a correlation between the lambdas at the different levels? What is the rationale for not having identical values across levels?

- Table 1: effect size (in ms) should be provided to allow comparison between actual and predicted RT effects.

Reviewer #3: In “Tracking human skill learning with a hierarchical Bayesian sequence model” Eltetö and colleagues apply a hierarchical sequence model to reaction time data from an alternating serial response time task. The hierarchical Chinese restaurant process (HCRP) with each level of the hierarchy coding for sequences of a particular length is enriched with a process of forgetting (forgetful HCRP). The model is then applied to data from 25 participants who completed 10 runs with a total of more than 20000 trials. The first 8 runs were generated by the same statistical trigram structure, while sessions 9 and 10 where used to probe the stability of the learned sequences against novel trigrams. In combination with some low level feature such as repetition and error-related effects, the model was able to capture how participants learned the sequence. The success fitting is illustrated by the correlation of predicted traces to left out data. The fitted parameters capture the nature of the task (over session) and, importantly, correlate with working memory indices acquired with classical working memory tasks.

This is a very nice application of a model for implicit learning tested on a rich dataset that has enough trials (25 participants, >20000 trials each) to actually allow to fit the model and investigate effects of changing structure in the task. Both task and model are well explained. I have few comments and questions on the model and model fitting to the authors (see below).

Jakob Heinzle

Major:

Model fitting: It was not entirely clear to me how ABC was applied, here. You describe that you simulated five instances of new seatings in every trial and then averaged over them. Is this enough to get stable fits? How could you assure this, other than by looking at the correlation to the held-out data? In addition, it was not clear to me how you dealt with the held out data. Did you fit the model up to the time of the held-out data and then modeled the held out data to check the posterior predictive value? Or was the model fit on the entire data/session and then tested on the held-out data? Did you restart the sequence for learning after the held-out data, or how did you include the sequence dependencies across the boarder of the held out data?

Discussion: While you elaborate in the discussion on possible applications e.g. in sequence learning in song birds, I was missing a discussion on how you think the fHCRP in combination with ABC could be implemented in the brain. It would be interesting to read your thoughts on this as a generative model of behavior should neuronally implemented as well.

Availability of data and model: While you say that the relevant data will be made available, it is not clear whether this includes the raw data and the full analysis/modeling code. You have acquired a unique data set which could serve other groups as a basis for their modeling. Please mention the availability of the data within the manuscipt.

Minor:

Figure 1: The 95% CI are not visible. Are these confidence intervals of the mean. Could show standard deviations instead to increase visibility.

Figure 5: It was not clear to me whether you applied any correction for multiple comparison for the multiple WM tests and levels of alpha and lambda? Given that you suggest that the detailed model is necessary to extract parameters that relate implicit sequence learning with WM performance, it would be good to state this clearly. In particular, because other studies have suggested there is no relation as you discuss.

Figure 6c: It would be interesting to discuss not only the trace of sequence learning but also others. Repetition, for example, shows an increase for the last two sessions. What ist the interpretation of this? It is also not clear, what exactly the plotted curves for individual regressors show? Unique variance?

Priors: When fitting the model with the forgetful prior (Figure 8). Was the posterior value of lambda always at the lowest end of the uniform prior? If so, why did you chose exactly that value? Where were the gaussian priors in sessions 2-10 truncated? At the border of the uniform prior in session 1? This was not clear to me.

**Have the authors made all data and (if applicable) computational code underlying the findings in their manuscript fully available?**

Reviewer #1: **No: **

Reviewer #2: **No: **Not only the code and data were not provided at this stage, the manuscript is lacking a detailed description of the model. As mentioned in my comment, this omission needs to be addressed in future revisions.

Reviewer #3: Yes

PLOS authors have the option to publish the peer review history of their article (what does this mean?). If published, this will include your full peer review and any attached files.

Reviewer #1: **Yes: **JEAN DAUNIZEAU

Reviewer #2: No

Reviewer #3: **Yes: **Jakob Heinzle
---

## [Decision Letter · Decision Letter 1]

10 Oct 2022

Dear Ms Éltető,

Thank you very much for submitting your manuscript "Tracking human skill learning with a hierarchical Bayesian sequence model" for consideration at PLOS Computational Biology. As with all papers reviewed by the journal, your manuscript was reviewed by members of the editorial board and by several independent reviewers. The reviewers appreciated the attention to an important topic. Based on the reviews, we are likely to accept this manuscript for publication, providing that you modify the manuscript according to the review recommendations.

Sincerely,

Christoph Mathys

Academic Editor

PLOS Computational Biology

Samuel Gershman

Section Editor

PLOS Computational Biology

Reviewer's Responses to Questions

**Comments to the Authors:**

Reviewer #1: I woud like to congratulate the authors for their thorough responses and revisions (addiitonal minor comments I had were also answered when adressing the comments of the other reviewer).

I think this work will provide a significant contribution to the field!

Jean Daunizeau.

Reviewer #2: The authors answered all my concerns. I appreciate in particular the additional model comparison and algorithm explanation. I still found the provided code a bit lacking (I expected a README and comments structuring the code: the repo is so far quite bare and unusable without some serious guessing work). I trust the authors to address this last point before publication. I otherwise have no further comments.

Reviewer #3: Comments on Revised manuscript “Tracking human skill learning with a hierarchical Bayesian sequence model”.

In this revision, the authors have addressed my concerns in a satisfactory manner. I have two comments remaining directly related to their answers.

Jakob Heinzle

Major:

I was a bit surprised about your statement that the data was already published in the recently accepted paper by Török et al. (PLoS Comput Biol 18(6):e1010182. https://doi.org/10.1371/journal.pcbi.1010182). I understand this is exactly the same data that you analyse, here. I had not realized this when reading the initial manuscript, and I think it should be made more transparent. While you cited the Török study (a preprint version of it) for additional reference on the methods, I could not find a statement that it is indeed the same data that you are using. I consider it as fundamentally important to mention in your paper that it is not the first time this data are published. Readers need to understand that you present a novel modeling analysis, but of an existing data set. Also, in the discussion, you should mention that the results of Török et al rest on exactly the same data. This is important for the questions regarding model comparison that other reviewers have brought up and which I think are highly relevant. Note: I added this point as major, not because I think that it will entail a lot of work, but because of its importance.

Minor:

There are still some things unclear about ABC. E.g. what stopping criterium do you use, also, it is clear that using 5 instantiations of the CRP is not reaching a plateau. Why is 5 still a good number? In addition, I think you need to mention more clearly that your held out dataset is not fully independent. I understand that you need to include the trials of the middle segment (the held out data) in your CRP updates to give the model the right context for the later trials. However, this means that your parameter estimates are conditional on the inputs (“features”) of the training set, even if you do not use the RT measurement of that period to fit the hyperparameters. In machine learning, one would usually not include the features of the test set in the training, even if test labels are not used. I realize that this is a tricky point and, hence, I think it is important that you explain this deviation from an ideal held-out set to the reader.

**Have the authors made all data and (if applicable) computational code underlying the findings in their manuscript fully available?**

Reviewer #1: Yes

Reviewer #2: Yes

Reviewer #3: Yes

PLOS authors have the option to publish the peer review history of their article (what does this mean?). If published, this will include your full peer review and any attached files.

Reviewer #1: **Yes: **Jean Daunizeau

Reviewer #2: No

Reviewer #3: **Yes: **Jakob Heinzle

Figure Files:

Data Requirements:

Reproducibility:

References:

---

## [Decision Letter · Decision Letter 2]

31 Oct 2022

Dear Ms Éltető,

We are pleased to inform you that your manuscript 'Tracking human skill learning with a hierarchical Bayesian sequence model' has been provisionally accepted for publication in PLOS Computational Biology.

Best regards,

Christoph Mathys

Academic Editor

PLOS Computational Biology

Samuel Gershman

Section Editor

PLOS Computational Biology

Reviewer's Responses to Questions

**Comments to the Authors:**

Reviewer #3: The authors answered all my questions and I have no further concerns.

Congratulations on this nice piece of work.

Jakob Heinzle

**Have the authors made all data and (if applicable) computational code underlying the findings in their manuscript fully available?**

Reviewer #3: None

PLOS authors have the option to publish the peer review history of their article (what does this mean?). If published, this will include your full peer review and any attached files.

Reviewer #3: **Yes: **Jakob Heinzle

---

## [Editor Report · Acceptance letter]

16 Nov 2022

PCOMPBIOL-D-22-00132R2 

Tracking human skill learning with a hierarchical Bayesian sequence model

Dear Dr Éltető,

I am pleased to inform you that your manuscript has been formally accepted for publication in PLOS Computational Biology. Your manuscript is now with our production department and you will be notified of the publication date in due course.

With kind regards,

Anita Estes
